# Logical design of synthetic *cis*-regulatory DNA for genetic tracing of cell identities and state changes

Carlos Company[1,5], Matthias Jürgen Schmitt[1,5], Yuliia Dramaretska[1,5], Michela Serresi [1,5], Sonia Kertalli[1], Ben Jiang [1], Jiang-An Yin [2], Adriano Aguzzi [2], Iros Barozzi [3,4] & Gaetano Gargiulo [1] ✉

Descriptive data are rapidly expanding in biomedical research. Instead, functional validation methods with sufficient complexity remain underdeveloped. Transcriptional reporters allow experimental characterization and manipulation of developmental and disease cell states, but their design lacks flexibility. Here, we report logical design of synthetic *cis*-regulatory DNA (LSD), a computational framework leveraging phenotypic biomarkers and trans-regulatory networks as input to design reporters marking the activity of selected cellular states and pathways. LSD uses bulk or single-cell biomarkers and a reference genome or custom *cis*-regulatory DNA datasets with user-defined boundary regions. By benchmarking validated reporters, we integrate LSD with a computational ranking of phenotypic specificity of putative *cis*-regulatory DNA. Experimentally, LSD-designed reporters targeting a wide range of cell states are functional without minimal promoters. Applied to broadly expressed genes from human and mouse tissues, LSD generates functional housekeeper-like sLCRs compatible with size constraints of AAV vectors for gene therapy applications. A mesenchymal glioblastoma reporter designed by LSD outperforms previously validated ones and canonical cell surface markers. In genome-scale CRISPRa screens, LSD facilitates the discovery of known and novel bona fide cell-state drivers. Thus, LSD captures core principles of *cis*-regulation and is broadly applicable to studying complex cell states and mechanisms of transcriptional regulation.

The precise identification of specific cell types and transient states is essential to understanding biological processes in which a diverse set of cell types/states contribute to tissue homeostasis. Accurately defining cell entities, states, as well as boundaries and trajectories governing physiological and pathological transitions, is particularly important for understanding cell responses to complex alterations of homeostasis, such as cancer[1–3] and viral infections[4]. Moreover, monitoring the spatiotemporal activation of a given pathway is instrumental to dissecting the underlying biology as well as monitoring the response to biological and chemical perturbations. While single-cell genomics and proteomics are providing increasingly powerful multi-omics maps of cellular processes in steady-state conditions and in

[1]Max-Delbrück-Center for Molecular Medicine in the Helmholtz Association (MDC), Robert-Rössle-Str. 10, 13092 Berlin, Germany. [2]Institute of Neuropathology, University Hospital Zurich, University of Zurich, 8091 Zurich, Switzerland. [3]Center for Cancer Research, Medical University of Vienna, Borschkegasse 8a, 1090 Vienna, Austria. [4]Department of Surgery and Cancer, Imperial College London, London, UK. [5]These authors contributed equally: Carlos Company, Matthias Jürgen Schmitt, Yuliia Dramaretska, Michela Serresi. ✉e-mail: gaetano.gargiulo@mdc-berlin.de

longitudinal analyses, equally comprehensive experimental tools to trace live cells are lagging behind. Lineage tracing in developmental settings exploits the genetic tagging of a single gene to map the fate of phenotypes associated with the expression of the selected gene[5]. Limitations associated with engineering an endogenous locus with a reporter include the assumption that gene expression regulation of the selected biomarker is a direct proxy of the phenotype of interest. This may not be systematically warranted when complex regulatory networks are studied. Conversely, selecting *cis*-regulatory elements to design synthetic cassettes showing sufficient specificity is complicated by our incomplete functional annotation and the mechanistic understanding of *cis*-regulation for most genes.

Synthetic transcriptional reporters may be assembled by juxtaposing candidate *cis*-regulatory DNA sequences. In cellular and molecular genetics, designing synthetic reporters starting from naturally occurring *cis*-regulatory elements responsive to well-defined signaling pathways or to combinations of transcription factors is a well-established strategy[6,7]. Significant effort was directed towards generating and selecting synthetic reporters using massively parallel sequencing or mixed computational design strategies[8–12]. This revealed the promising potential of this approach, as well as the limitations associated with incomplete control over the design, which remains challenging[9,11]. Importantly, how biochemically defined endogenous *cis*-regulatory DNA informs the generation of synthetic enhancers remains unclear[13]. As a rule of thumb, success in generating functional reporters is dramatically increased by combining candidate enhancers with validated *cis*-regulatory elements, such as viral or the β-globin minimal promoter. This, however, holds undefined consequences for the specificity and sensitivity of such reporters.

We recently developed a method to generate synthetic transcriptional reporters for genetic tracing (termed synthetic locus control regions, or sLCRs) and used these to study the significance of glioblastoma heterogeneity in vitro and in vivo[14]. This method is potentially applicable to a variety of biological settings when a more streamlined, automated implementation of computational workflow is implemented. Here, we present a computational framework to de novo assemble functional sLCRs capable of working on stereotyped inputs and returning an optimal candidate sLCR output. Logical design of synthetic *cis*-regulatory DNA (or LSD) generates one candidate sLCR from a user-dependent list of biomarkers and transcription factors by performing an unbiased search for optimal *cis*-regulatory DNA within the reference genome or a user-defined set of candidate *cis*-regulatory elements. We complement LSD with a computational approach to rank candidates with naturally occurring or synthetic DNA based on the signal-to-noise ratio of a phenotype of interest. In turn, we benchmarked LSD's performance using validated reporters and offered a proof-of-concept on how to exploit LSD towards the systematic characterization of cell types and states as well as to validate bulk and single-cell genomic studies.

## Results

### Logical design of synthetic *cis*-regulatory DNA

To make the design of sLCRs robust and generally applicable, we developed a fully automated framework that couples the selection of putative phenotype-specific *cis*-regulatory elements (CREs) to an iterative ranking in descending order of phenotypic representation.

The pioneering computational framework, termed logical design of synthetic *cis*-regulatory DNA or LSD, uses two inputs: (1) a list of signature genes, which are biomarkers representative of the target phenotype, and (2) a list of transcription factors (TF) with known DNA-binding motifs potentially regulating such genes (Methods). Both lists can be based on differential or absolute gene expression, but in principle, they could be defined based on any set of criteria (Fig. 1a). Building on our first-generation sLCR design algorithm[14], LSD first scans the regulatory landscapes of the signature genes to predict

putative *cis*-regulatory elements (CREs) regulating them (Fig. 1a and Methods). By default, the boundaries of these regulatory landscapes are defined using annotated CTCF binding sites[15,16] flanking the signature genes. Such a 'nearest CTCF neighbor' criterion conservatively approximates the functional definition of chromatin loops[17] and topologically associated domains[18]. Alternatively, users can impose experimentally defined boundary regions, including from ChIP-seq data for other structural DNA-binding proteins and chromosome conformation capture experiments (see below). LSD scans these regulatory landscapes using a 150 bp sliding window, in 50 bp steps. This process returns a pool of putative CREs that are scored and ranked using the set of TF-binding models defined by the user. The scoring uses the following criteria: (I) the absolute number of TFBS; (II) the diversity of the TFs showing high affinity for the regions; and (III) the distance from the nearest endogenous transcriptional start site (TSS) (Fig. 1a and Methods). Candidate sLCRs are then generated from these CREs. The goal is to include the smallest number of CREs that faithfully represent the complexity of the regulatory inputs. To do so, LSD iteratively ranks the highest-scoring CREs until all pre-defined TFs showing at least one binding site are represented (Fig. 1a and Methods). The output of LSD is proportional to the number of input TFs, the number of signature genes and the size of the genomic loci containing these genes.

While our implementation relies on the scheme and assumptions outlined above, our dynamic strategy can be complemented or replaced by one based on a different set of rules or defined by hypothesis-driven criteria, such as focusing on user-defined TFBS representation.

To directly compare reporters generated through the first-generation approach[14] to those designed with LSD, we designed three original reporters for recurrent glioblastoma expression subtypes. The PNGT3, CLGT3 and MGT4 sLCRs were designed in an unbiased manner by LSD and their specific signature genes were identical to those of the first-generation sLCRs (Supplementary Fig. S1a, b), while we defined the TF lists by differential enrichment (see Methods). A minor variation in the TFBS list permits the design of different reporters to potentially target the same phenotype. Single-sample gene set enrichment analysis (ssGSEA) of either TF list in TCGA GBM RNA-seq data showed that they are both representatives of their target phenotype (Fig. 1b and Supplementary Dataset S1). Of note, each reporter significantly enriched the TFBS sites specific for the intended glioblastoma subtype (Fig. 1c and Supplementary Fig. S1c, d). This indicates that LSD maintains the robustness of the original approach while operating fully automated. Importantly, hierarchical clustering of TFBS enrichment in reporters generated by either algorithm showed that the mesenchymal glioblastoma-subtype sLCR designed by LSD clustered with previously validated mesenchymal reporters in all the tested analyses (Fig. 1d and Supplementary Fig. S1).

### LSD allows for designing functional and specific sLCRs

To assess the performance of the LSD method, we next synthesized the LSD-designed mesenchymal glioblastoma sLCR (hereupon 'MGT4') and tested it head-to-head against first-generation MGT1-2 sLCRs constructed by user-supervised assembly and experimentally validated in vitro and in vivo[14]. Multiple sequence alignments of the three sLCRs show only one instance of conserved positional overlap for contiguous nucleotides of the size of a TFBS (>5 bp; Fig. 2a). This suggests that the TFBS grammar is sLCR-specific, despite all reporters targeting the same phenotype through transcription factor binding. Nevertheless, FACS analysis showed that MGT1 (first generation) and MGT4 (LSD) are similarly responsive to TNF-alpha, a driver of mesenchymal commitment in GBM cells and of MGT1-2 activation[14] (Fig. 2b). Of note, MGT1 exhibits a comparable expression regardless of whether genome engineering relies on lentiviral- or PiggyBac-based systems (Fig. 2b). This indicates that sLCRs' activity is mainly directed

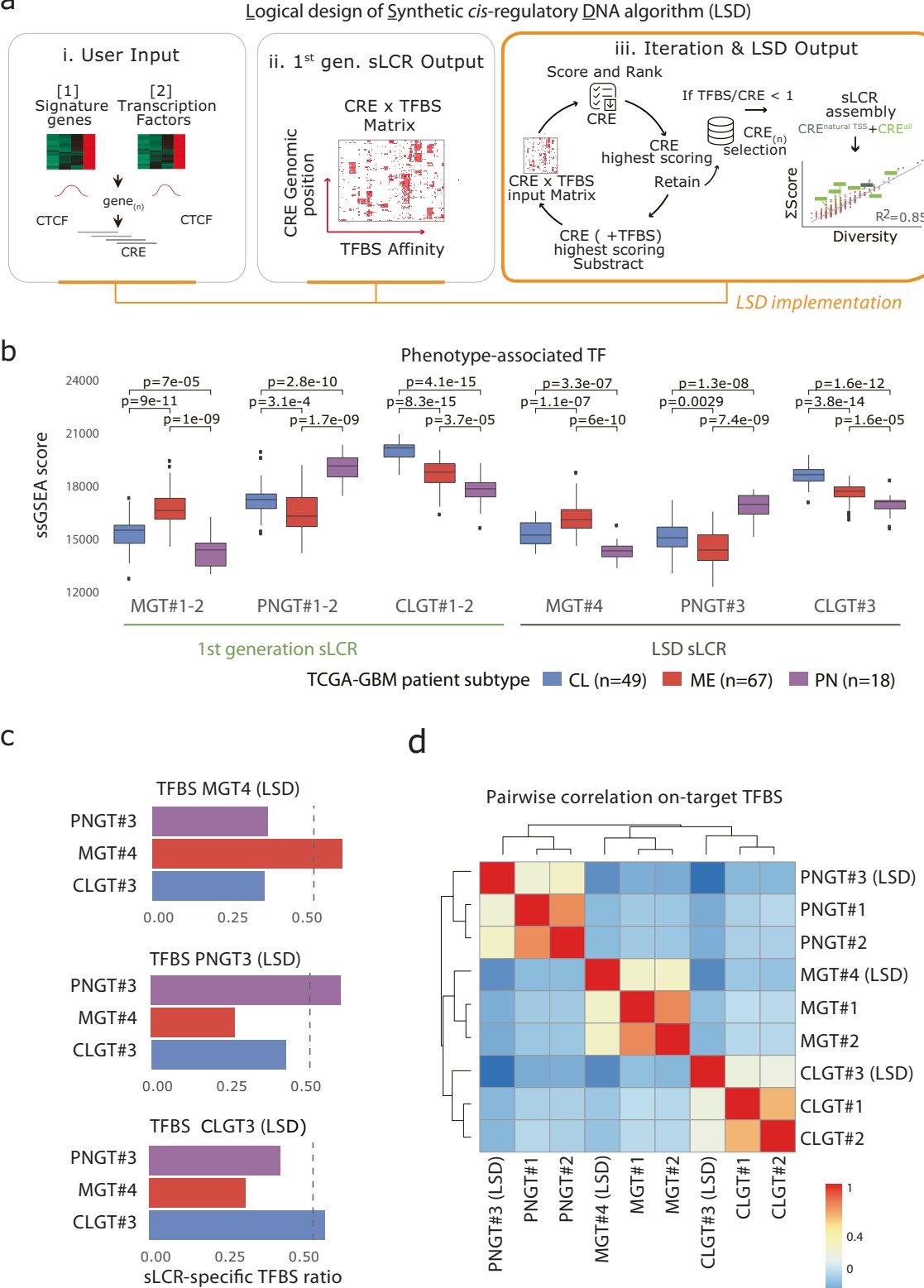

by the synthetic *cis*-regulatory DNA and largely independent of the genome integration bias of the vector system employed.

In recent studies, surface markers such as CD24 and CD44 have been used as experimental surrogates for opposite cell states in human glioblastoma (proneural/NPC-like and mesenchymal[19]) as well as in ovarian cancer cells (epithelial-mesenchymal[20]). To compare the performance of LSD-designed reporters with these established markers,

we conducted correlation analyses between module scores for PNGT3, CLGT3, and MGT4 input signature gene sets and module scores for major glioblastoma cell state gene sets[19,21–23], as well as individual CD24 and CD44 expression (Methods). In silico, leveraging a comprehensive pan-glioblastoma single-cell integration[24], we observed that PNGT3 correlated with the proneural/OPC-NPC meta-module, while MGT4 correlated with the mesenchymal meta-module (Fig. 2c and

**Fig. 1 | LSD streamlines the design of sLCRs from defined inputs. a** Schematic depiction of the LSD pipeline: from input signature genes and TFBS lists (i) it generates a CRE × TFBS matrix (ii; see Methods) and performs iterative selection of the top-ranked CREs (iii). Each iteration removes the highest scoring CRE and TFBS from the CRE × TFBS matrix until the CRE × TFBS contained no TFBS or CRE. The output of LSD is a ranked list of n CREs. The CRE closest to a natural TSS is prioritized. The example to the right illustrates a linear relationship between TFBS affinity and TFBS diversity for all CREs in the CRE × TBFS matrix (red circles; R2 = 0.86). In light green boxes, LSD ranked n=7 CREs (1050bp) covering >60% of the TFBS diversity. The TSS-containing CRE is in dark green. **b** Boxplot showing ssGSEA scores of The Cancer Genome Atlas Glioblastoma (TCGA-GBM) patient cohort for subtype-specific TF input lists. Each annotated GBM transcriptional subtype (CL – Classical, blue, n = 49; MES – Mesenchymal, red, n = 67; PN – Proneural, purple, n = 18) features statistical comparisons by two-sided pairwise t-test. Data distribution is shown, with box indicating the interquartile range and inner line indicating the median. Whiskers extend to represent the data range, including outliers. **c** Barplot showing the coverage of sLCR-specific TFBS lists (color) relative to the indicated TFBS input list (above). The dashed line denotes a threshold of 50%. **d** Heatmap showing the Pearson correlation between the TFBS score/diversity for each sLCRs-input TF list. sLCR synthetic locus control region, LSD logical synthetic cis-regulatory DNA, TF Transcription Factor, TFBS Transcription Factor binding site, CTCF CCCTC-binding factor, CRE cis-regulatory element, TSS Transcription Start Site, ssGSEA single sample gene set enrichment analysis. Source data are provided in the Source Data file.

Supplementary Fig. S2e). Notably, the overall correlation between patient-derived cell state signatures and our sLCR-based approach, which captures the entirety of the signature rather than relying on a single gene approximation, shows performance comparable to or superior to the use of single marker genes alone for inferring cell fate (Fig. 2c and Supplementary Fig. S2e), as gauged by the higher correlation between PNGT3 and proneural/NPC-like than CD24 (Fig. 2c and Supplementary Fig. S2e). This supports the ability of LSD sLCRs to recapitulate targeted identities and serve as a tool for tracing classical/AC-like modules where surrogate surface markers are unavailable (Fig. 2c and Supplementary Fig. S3f). In vitro, MGT4 expression accurately reflects a well-established model of TNF-driven PMT, whereas CD24 and CD44 do not (Fig. 2d), demonstrating the superior experimental performance of sLCRs in capturing comprehensive regulatory networks.

Transcriptional reporters typically consist of candidate enhancers upstream of a minimal promoter[11,25]. The use of non-specific promoters limits the phenotypic specificity of the reporter. In contrast, no minimal promoters were required to design functional and specific MES GBM sLCRs. To extend this observation, we next used LSD to systematically design sLCRs for a wide range of cell states and pathways' activation. These include the proneural and classical glioblastoma expression subtypes, an astrocyte-like glioblastoma cell state, ER-stress response, senescence, T-cell exhaustion, disease-associated microglia activation, and epithelial cells' response to SARS-CoV-2 infection. Consequently, the source of signature genes, transcription factors, the choice of reporter genes, an LSD-sLCR-independent selection marker, and vector backbones differed according to the intended outcome. As a result, we generated a broad pool of sLCRs whose connecting thread is being designed by LSD (Supplementary Dataset S1 and Methods). With such a diverse range of sLCRs, we were able to determine if LSD systematically generates functional reporters. Thus, we synthesized LSD-sLCRs and episomally tested their expression side-by-side with the experimentally validated first-generation reporters in three different cell lines. We transfected 28 different sLCRs into human epithelial HEK293T, mouse mesenchymal NIH3T3, and Chinese hamster ovary (CHO-1) epithelial cells to cover a minimal set of variables that allow assessing transcriptional competence and specificity, including developmental stage, tissue-specificity, and species-specificity. Despite the use of phenotype-agnostic cell types, upon transfection, the median expression of the sLCRs was significantly (p-value < 1e−10) distinct from the background (Fig. 2e). There was no obvious bias associated with the algorithm used to design them or the target phenotype, but the sLCR showed distinct patterns of expression in the three cell types. Overall, reporters designed with the human genome as a reference displayed a mild but statistically higher (p = 1.51e−05) expression in human cells if compared to mouse and hamster cell lines (Fig. 2e). Some LSD-sLCRs were marginally transcribed despite high transfection efficiencies, as gauged by the expression-independent fluorophore in all the tested lines (Supplementary Fig. S2b). This could be interpreted as either a measure of on-target specificity or a lack of transcriptional

competence. As exemplified by the case of our mesenchymal GBM sLCRs, those reporters were highly induced by TNF-alpha in GBM cells, confirming that they are functional but lowly expressed in transient transfection of non-glioblastoma cells, supporting their specificity (Fig. 2b and Supplementary Fig. S2b). Interestingly, one sLCR scoring very high on-target activity, such as PNGT2 in proneural human glioblastoma initiating cells, was well expressed in human 293T cells but less expressed in non-human cells (Supplementary Fig. S2a), suggesting that the species discordance might affect the output reporter activity.

Thus, LSD generates reporters whose specificity is linked to the source of signature genes, transcription factors and cell model systems, while its transcriptional competence is independent of minimal non-specific promoters.

## Benchmarking LSD by cis-regulatory score ranking towards defining basic principles of sLCR design

Given that the LSD approach can optimize TFBS complexity within selected CREs, we next sought to exploit glioblastoma first-generation validated MGT1-2 reporters[14] and proneural-to-mesenchymal transition (PMT) as a benchmark to predict the functionality and specificity of newly LSD-designed sLCRs.

First, we set out to estimate an indicative number of distinct TFs that should be represented by cognate TFBS in a given sLCR in order for it to be functional and specific. To this end, we set out to determine background TFBS complexity by using different randomization strategies. First, we sampled TFs from the overall pool of database-annotated human TFs while maintaining the signature genes constant (termed 'Random TF'; Fig. 3a). Second, we randomly selected matched-sized sets of input genes from the human genome (GRCh37/hg19) and maintained the MGT1-2 TF list (termed 'Random Sig.-TF'; Fig. 3a). We then calculated, for each of the two scenarios and for increasing numbers of sampled TFs, a measure of specificity, defined as the fraction of TF-genes included in the designed sLCR, which we annotated as MGT factors. This fraction increased as a function of the input number of random TFs, and this trend was more marked when the MGT1-2-4 signature genes were used (Fig. 3a). Importantly, all mesenchymal sLCRs (MGT1-2-4), which were designed with less than one hundred TFs, covered >50% of the entire TF repertoire, and the observed/expected ratio was superior to both the background and the phenotypically distinct classical and proneural reporters (Fig. 3a). Interestingly, despite the fact that the MGT4 reporter was designed by LSD on a different TFBS list, it outperformed the first-generation MGT1-2 on their specific TFBS input list (Fig. 3a). Likewise, LSD-designed sLCRs PNGT3 and CLGT3 always outperformed 1st generation and phenotypically distant reporters (Supplementary Fig. S3a–c). In fact, using their specific signature and TF-gene inputs, they all showed an observed/expected TFBS ratio above both background and other functional reporters designed to have specificity for a different phenotype (Supplementary Fig. S3a). Hence, the LSD-sLCR approach outperformed the supervised selection by enriching the number of

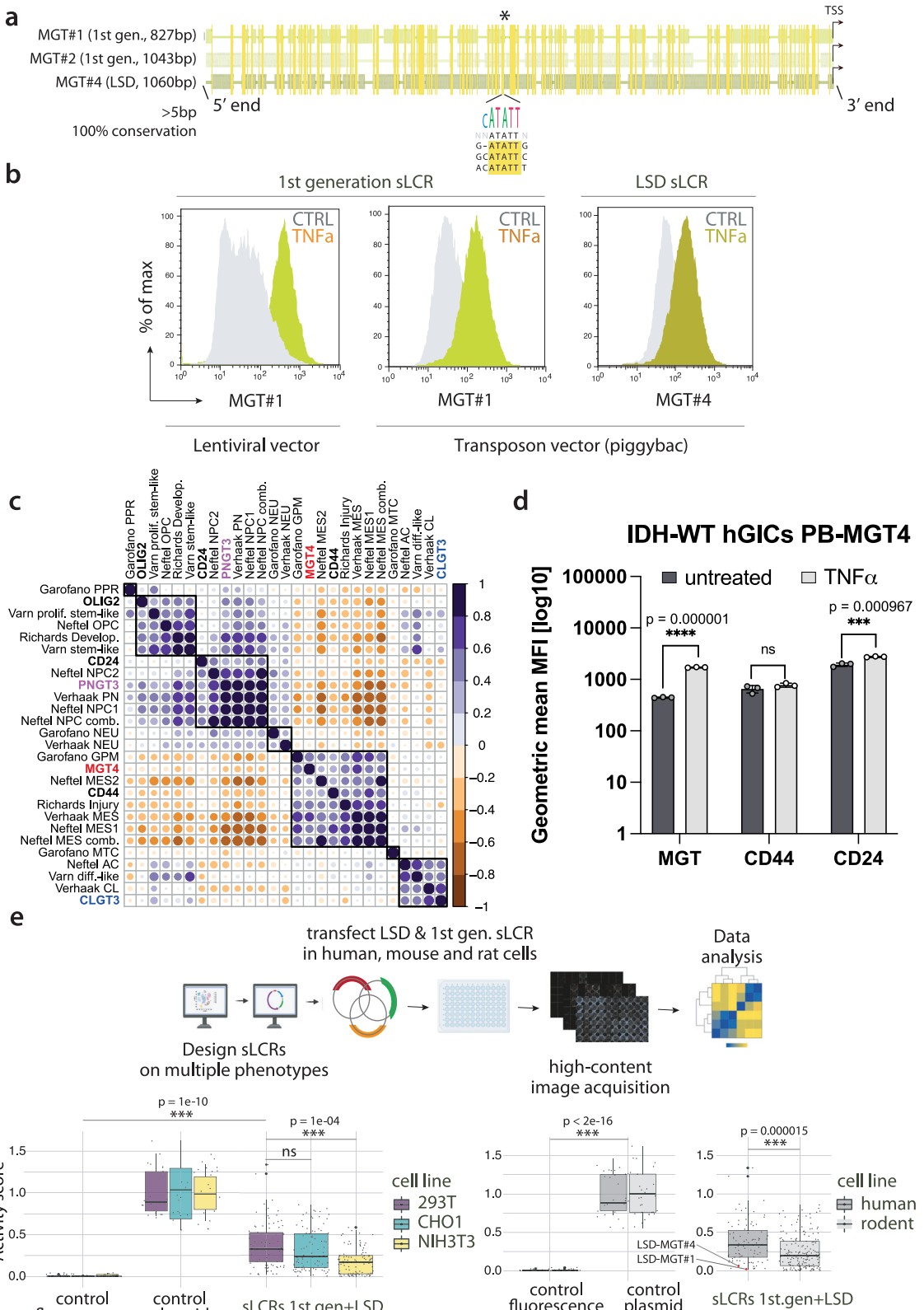

TFBS detected above the signal-to-noise ratio, even at large TF numbers.

Beyond the threshold of one hundred TFs, our randomization approach suggests that while the number of target TFBS proportionally increases with the size of the sLCR, this does not affect the fraction of specific TFs recovered (Fig. 3a). This prediction is compatible with the idea that unnecessary long sLCRs are prone to unpredictable

regulatory activities, suggesting that the length of an sLCR is a critical determinant of its specificity and sensitivity. Thus, we next set out to define a minimal number of individual CREs that would predict a functional sLCR while aiming at maximal TFBS potential. To this end, we quantified the marginal information gain (i.e., the number of distinct TFs included in the sLCR, as quantified by the cognate TFBS) of increasing the number of individual CREs to be included in an sLCR.

**Fig. 2 | LSD allows the design of functional and specific sLCRs. a** Multiple sequence alignment (see Methods) of the 1st generation MGT1 and MGT2 and the LSD MGT4 reporters. The conserved positional overlap is denoted by the asterisk and graphically represented by the sequence logo. **b** FACS analysis of MGT1 (left & center) and MGT4 (right) sLCRs expression in human glioblastoma-initiating cells with (lime) or without (gray) TNFa stimulation. Note the similar induction between lentiviral- and transposon-engineered cells for 1st generation MGT1 sLCR, and between MGT1 and LSD-designed MGT4. **c** Correlation plot between patient-derived glioblastoma cellular state signatures and module scores of sLCR signature genes for pan-GBM data from Ruiz-Moreno et al.[24]. Purple denotes positive, whereas orange indicates negative correlation and dot size represents associated p-value. **d** Bar-plot quantification of FACS data showing mean MGT4 expression and CD44/CD24 staining intensity with (gray) or without (black) 48h Tumor Necrosis-factor alpha (TNFa) treatment (10ng/ml) in IDH-wildtype human glioma-initiating cell (hGICs). Data are presented as mean +/− standard deviation. Statistical significance values were calculated by two-sided unpaired t-test. Error bars denote standard deviation (n = 3 biologically independent samples). **e** Above, schematic depiction of the systematic screening of sLCRs designed on diverse phenotypic signatures in three different species (partially assembled using BioRender.com). Lower left, box plot of indicated sLCRs (n = 28) transfected in human epithelioid 293T (purple), hamster epithelioid CHO-K1 (teal) and mouse fibroblastoid NIH3T3 (yellow) cell lines. The X-axis shows fluorescence normalized by controls and transfection efficiency per cell line. Each sLCR measurement was assessed in technical replica (n = 3). Left, positive (n = 5) and negative controls (n = 3) denote CFP, GFP, mCherry and iRFP670 expression driven by non-sLCR promoters and fluorescence background in each channel, respectively. Lower right, box plot shows relative activity of human sLCR transfected in human (293T;dark gray) or non-human (CHO-K1, NIH3T3; light gray) cells. Data distribution is shown, with box indicating the interquartile range and inner line indicating the median. Whiskers extend to represent the data range, including outliers. Statistics: 2-way ANOVA, followed by Dunnet post-hoc test. Source data are provided in the Source Data file.

We focused this analysis on thirteen experimentally tested sLCRs targeting various phenotypes. Fitting a model to the relationship between the TFBS fraction and the number of CREs included using LASSO (R > 0.9; Fig. 3b), retrospectively determined that the top 20% of output CREs is sufficient to represent >50% of the maximal regulatory potential in a given phenotype. For example, for the herein-validated MGT4, we selected 7 out of 24 CREs (29%), for a total length of the sLCR of 1060 bp, and an observed/expected ratio of 0.56. Taken together, the results of this analysis suggest that, if input sets with comparable sizes are used for sLCR design, merging 25–30% of the top-ranking CREs may maximize the chances of obtaining a functional sLCR with minimal size and thus ectopic activities (see Discussion).

Finally, we tested whether the validated reporters could inform a model to predict the phenotypic specificity of a given candidate *cis*-regulatory DNA sequence, endogenous or synthetic. To this end, we established a *cis*-regulatory score ranking. Such ranking is based on the correlation between a score summarizing the overall affinity of each sLCR to the phenotype-specific TFs (termed 'Qscore'), and a score proportional to the phenotype-specific expression of the genes in the corresponding input included in the final design of each sLCR (termed 'SignScore'; see Methods).

Using the TFs and input gene signatures employed to design the MGT1-2 reporters (intended as a validated proxy of the mesenchymal GBM phenotype), the sLCR based on the mesenchymal TFs and signatures (including MGT4) outranked all the remaining reporters (Fig. 3c). To test the specificity of this ranking strategy, we introduced reporters potentially marking phenotypes distant from those of GBM subtypes. We used LSD to design sLCRs to map amacrine cells starting from murine single-cell RNA-seq profiling[26] and compared these reporters to a set of validated synthetic reporters generated by various approaches to perform gene therapy in mouse retinal cell types[11]. Importantly, mouse retinal reporters outranked glioblastoma sLCRs in the amacrine phenotype ranking, while they sat at the bottom of the mesenchymal glioblastoma phenotype ranking (Fig. 3c, d). Likewise, when this analysis targeted classical or proneural GBM TFBS selections, their respective reporters outperformed all the others (Supplementary Fig. S3b, c).

Overall, by using our validated sLCRs as a benchmark, we set up a series of computational strategies that can aid in the design of functional reporters and measure the *cis*-regulatory potential affinity of synthetic and endogenous reporters to their target phenotype.

## LSD incorporates single-cell RNA-seq as signature gene input
Having established the empirical performance of LSD on bulk RNA-seq and reference genomes, we sought to exploit scRNA-seq data as signature gene inputs for LSD.

Since glioblastoma subtypes were recently reassessed as distinct cell states using single-cell RNA-seq[19], we next used these meta-signatures to generate sLCRs from scRNA-seq inputs. As a resource for glioblastoma-specific TFBS, we resorted to either bulk or scRNA-seq lists and designed the reporters for the four scRNA-seq glioblastoma states.

Whereas the mesenchymal glioblastoma subtypes and states are substantially overlapping, and ssGSEA analysis indicated that MGT1-2 (1st gen) and MGT4 (LSD) are already representative of this state (Supplementary Fig. S3d and ref. 14), while the relationship between non-mesenchymal glioblastoma subtypes and states is unclear. Computationally, proneural and classical sLCRs broadly span through two states, but the AC-like sLCRs clearly represent the classical GBM subtype (Supplementary Fig. S3d, e and ref. 14). Hence, the resource provided herein (Supplementary Dataset S1) may be helpful in defining critical cell-intrinsic signaling and cell fate changes upon perturbation, which may be particularly useful to study the significance of specific cell states.

## LSD integrates 3D contact maps and DNA accessible in chromatin as custom inputs
Chromatin accessibility is a primary determinant of cell type-specific *cis*-regulatory activity, and accessible TFBS are more likely to bind cognate trans-regulatory factors[27,28]. The 3D genome organization guides the spatiotemporal function of enhancers in the mammalian genome[17,18]. The increasing availability of cell type-, developmental state- and disease-specific chromatin structure maps prompted us to test whether sLCRs may be designed with the aid of such input datasets.

First, we designed the mesenchymal glioblastoma MGT4 LSD-sLCRs from four alternative input combinations (Fig. 4a). The nearest-neighbor CTCF binding sites approach applied to the full extent of the reference genome is the standard approach described above (Fig.4a, I). MGT4 design iterations were obtained by either restricting the reference genome to the cancer-specific accessible genome, as defined by ATAC-seq profiles[29] (Fig. 4a, III and IV), or by extending the space for candidate *cis*-regulatory domains to tissue-specific TADs[30] (Fig. 4a, II and IV). We used four independent sources of ATAC-seq data, including mouse regulatory regions[31]. The signature genes and TFBS input were identical to those used in MGT4 LSD-sLCR. Therefore, this analysis generates a set of distinct LSD-sLCR designs potentially targeting the same phenotype.

To compare the sLCRs designed by LSD according to the four different models (Fig. 4a), we constrained sLCR size to that of the validated MGT4 reference. We computed the *cis*-regulatory score affinity to the MGT4 target phenotype for all the mesenchymal GBM sLCRs designed with the above input iterations and for all other available reporters. Phenotypic score ranking shows that applying interactions to the input DNA changes the in silico specificity of the reporters (Fig. 4b). Yet, all mesenchymal reporters occupy a distinct

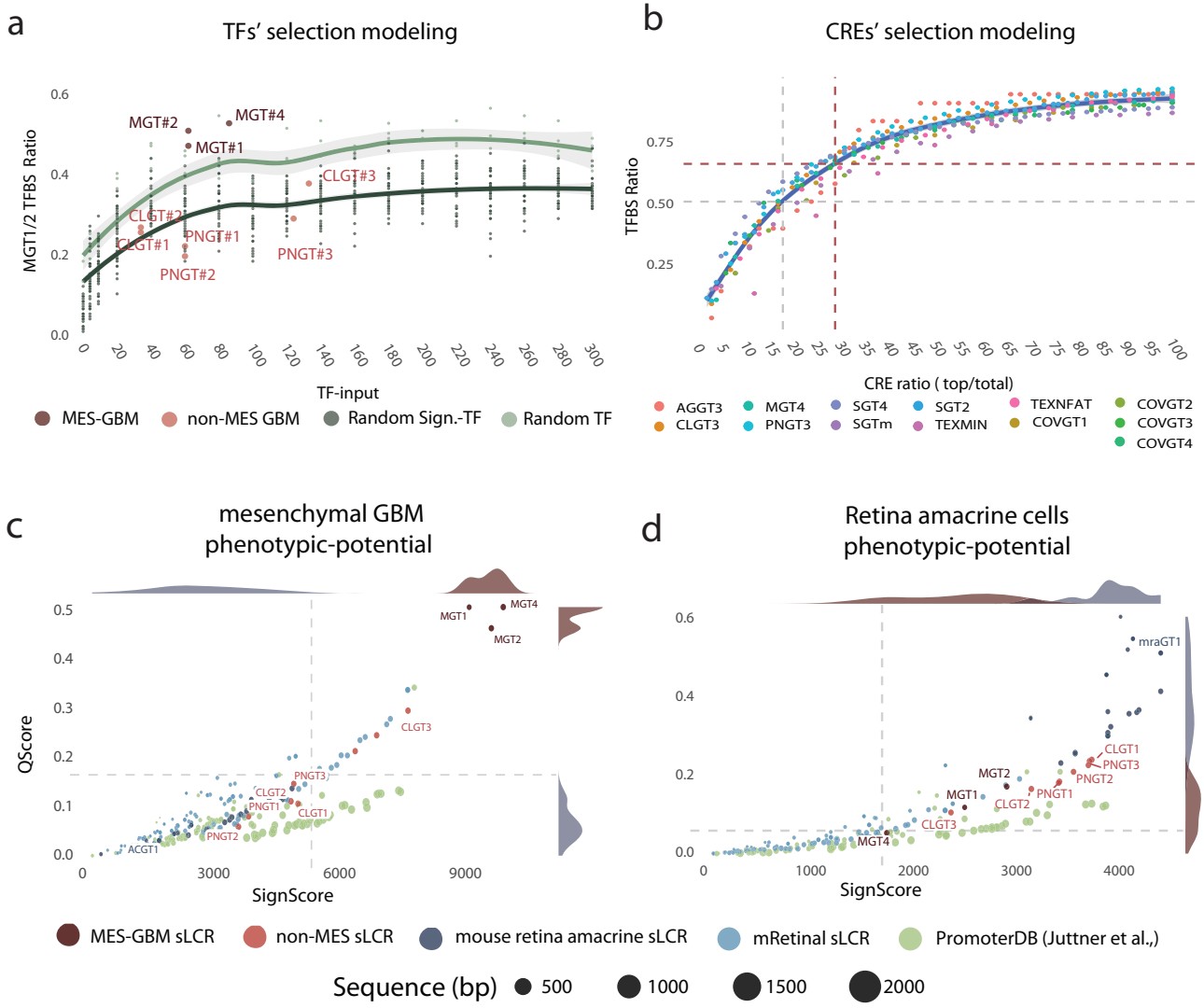

Fig. 3 | **Towards defining endogenous and synthetic reporters' phenotypic potential via TFBS enrichment ranking. a** Scatter plot showing the mesenchymal sLCRs TFBS affinity ratio for on-target, off-target and scrambled sLCRs. The Y-axis indicates the observed/expected ratio (i.e., MGT1-2 observed/input TFBS). The X-axis denotes the number of input TF. First-generation and LSD-sLCR are indicated. Scrambled sLCR were designed using LSD and input from random sampling of TFs from the general pool of annotated human TFs (random TF) or random selection of genes from the human genome (random Sign-TF). Fitted lines indicate LOESS regression with 95% confidence interval. **b** Scatter plot showing the TFBS affinity ratio as a function of increasing numbers of CREs. Values are calculated for each functional sLCR assessed experimentally (Fig. 2). Logarithmic regression was used to fit the curve. The gray dashed line indicates that the CRE ratio is >50% of TFBS with R2 = 0.96 and the blue solid line marks MGT4. **c** Scatter plot showing the signature score (x-axis) and affinity score (y-axis; see Methods) of the indicated reporters for the mesenchymal phenotype. Note the antagonistic phenotypic scoring of glioblastoma (reds) and neural retina amacrine cell reporters (blues). **d** Phenotypic scoring of the same reporters in **c** for a retina amacrine cell phenotype. sLCR synthetic locus control region, LSD logical synthetic cis-regulatory DNA, TF Transcription Factor, TFBS Transcription Factor binding site, CRE cis-regulatory element. Source data are provided in the Source Data file.

space from phenotypically distant sLCRs and background sequences, and the mesenchymal sLCR designed with the largest input for TFBS demonstrated a modest performance advantage over validated reporters using the standard inputs (Fig. 4c).

To complement the *cis*-regulatory ranking with a statistical approach that estimates data distributions with limited sampling and in the absence of major assumptions, we next used a bootstrapping analysis strategy[32]. We included all the different MGT4 iterations, non-mesenchymal sLCRs and phylogeny-distant DNA control regions from the SARS-CoV-2 viral genome. We conducted iterative random sampling of MGT4 TFBS enrichment at each LSD reporter under testing ($n = 1000$) using a fixed TFBS number (25% of the total, 231, with replacement). At each iteration, the total value of the random TFBS selection was calculated, thereby creating a distribution of TFBS enrichments for each LSD reporter. Then, each data distribution was compared against the others for each LSD reporter (greater = TRUE). The bootstrapping analysis established a hierarchy of significant pairwise correlations between distributions, in descending order. The significance directly correlated with the size of the input reference for TFBS search (Supplementary Fig. S4b, c). The number of significant events occurring decreased when the size of the ATAC-seq data was smaller than the reference genome and improved when the size was comparable (e.g. MIII; Fig. 4b, c and Supplementary Fig. S4d). In other words, constraining the TFBS search to a limited ATAC-seq dataset results in fewer significant interactions than when using the unrestricted reference genome. This can be interpreted as the limited likelihood of covering the entire TFBS repertoire in a short sLCR, which increases the number of CREs required to represent the target *cis*-regulatory potential (i.e., the overall size of the sLCR). Consistently, when LSD was restricted to the GBM ATAC-seq genome and the

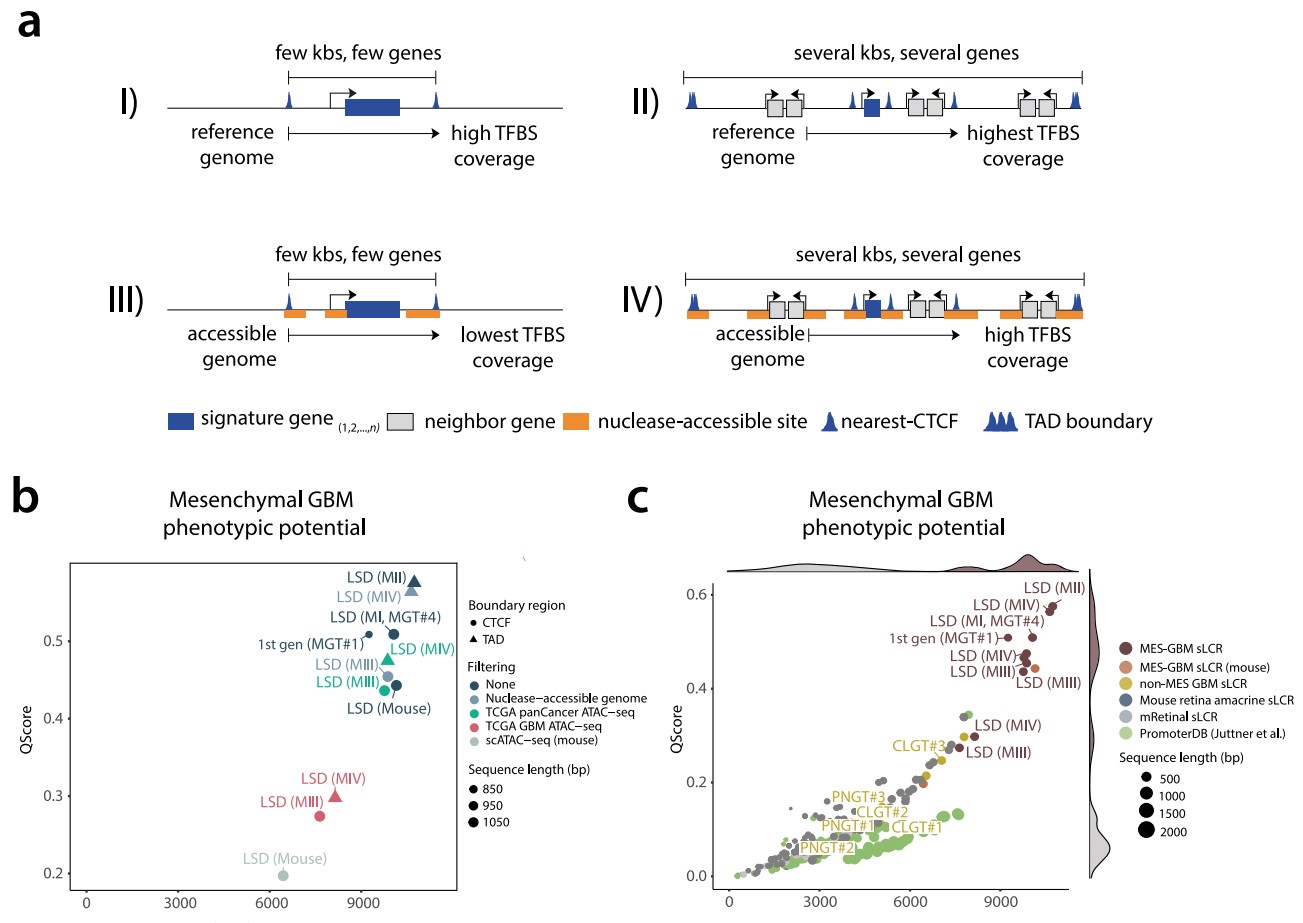

**Fig. 4 | Integrating chromatin accessibility and 3D contact maps as input for LSD. a** Schematic representation of alternative LSD input combination models. **b** Scatter plot showing the signature score (x-axis) and affinity score (y-axis; see Methods) of the indicated reporters for the mesenchymal phenotype. Different filtering methods are denoted by color codes and dot-size indicates sequence length in basepairs. Note the improved on-target score for a mesenchymal sLCR designed by LSD using model II (i.e. MGT5). **c** Scatter and density plots of mesenchymal reporters designed in **b** (dark red, light red, orange) with the addition of non-specific phenotypic reporters (blue, green, gray). Note that the mesenchymal phenotypic space is occupied by most mesenchymal reporters and that the including of accessibility and 3D contact data marginally increased or decreased sLCR scoring. sLCR synthetic locus control region, LSD logical synthetic *cis*-regulatory DNA, TFBS Transcription Factor binding site, TAD Topologically Associating Domain. Source data are provided in the Source Data file.

nearest-neighbor CTCF boundaries, the optimal sLCR designed by LSD outcome should be ~16% larger than MGT4. The use of larger *cis*-regulatory pools, such as the pan-cancer ATAC-seq or a curated set of nuclease-accessible genome regions from various ATAC-seq and DNAseI-seq maps, relieves such a constraint. In fact, the combination of TADs and the pan-cancer ATAC-seq datasets ranks similarly or slightly superior to MGT4 *cis*-regulatory potential, as gauged by both *cis*-regulatory score ranking and bootstrapping analyses (Fig. 4b, c and Supplementary Fig. S4b–d).

In conclusion, if the size of the custom *cis*-regulatory datasets for TFBS search is similar to the size of the unrestricted reference genome, LSD well integrates 3D contact maps and DNA accessible in chromatin inputs to extend the use of synthetic genetic tracing to the validation of functional experimental datasets.

## LSD enables designing of sLCRs compatible with the size constraints of AAV vectors

Adenoviral-associated vectors (AAV) are commonly used for delivering genetic material in vivo[11]. The applicability of AAV vectors is limited by the size, strength, and cell-selectivity of transgene expression driven by RNAPII promoters. Pioneering work in gene therapy identified the 511 bp phosphoglycerate kinase 1 (PGK) and the 232 bp short-version of the elongation factor-1 alpha (EFS) "housekeeping (HK)" gene

promoters as optimal drivers of gene expression in the mouse retina[33]. To test whether LSD enables the design of AAV-compatible short, potent and low-variable sLCRs, we defined a shortlist of HK signature and transcription factor genes (Fig. 5a, Supplementary Dataset S1 and Methods). To that end, highly expressed housekeeping signature genes were identified from Tabula Sapiens[34] and Tabula Muris[35] databases across human and mouse tissues (Supplementary Fig. S5a). The list of housekeeping transcription factor (TF) genes was curated by including previously published HK genes[36], since TFs often escape from capture by shallow scRNA-seq platforms. As input for LSD, we used four distinct iterations of human or human-mouse conserved signatures and TF genes to design four housekeeping genes genetic tracing sLCRs (HKGT1-4). Bona fide CREs were enriched using human TCGA pan-cancer ATAC-seq[29] (HKGT1-2) or mouse scATAC-seq[37] (HKGT3-4; Supplementary Dataset S1). From LSD output, the top two CREs were selected to generate a 300 bp HKGT-short sLCR comparable to EFS, and an additional CRE was added to create a 450 bp HKGT comparable in size to mouse and human PGK promoters. All HGKT were smaller than other well-known promoters such as UBC, CMV and EF1A (Fig. 5b). Phenotypic ranking based on the HK signatures showed that the sLCRs designed were distinct from well-established promoters like EFS, PGK, UBC, CMV, and EF1A (Supplementary Fig. S5b). Next, we synthesized 8 HKGTs sLCRs, as well as one EFS control, and generated

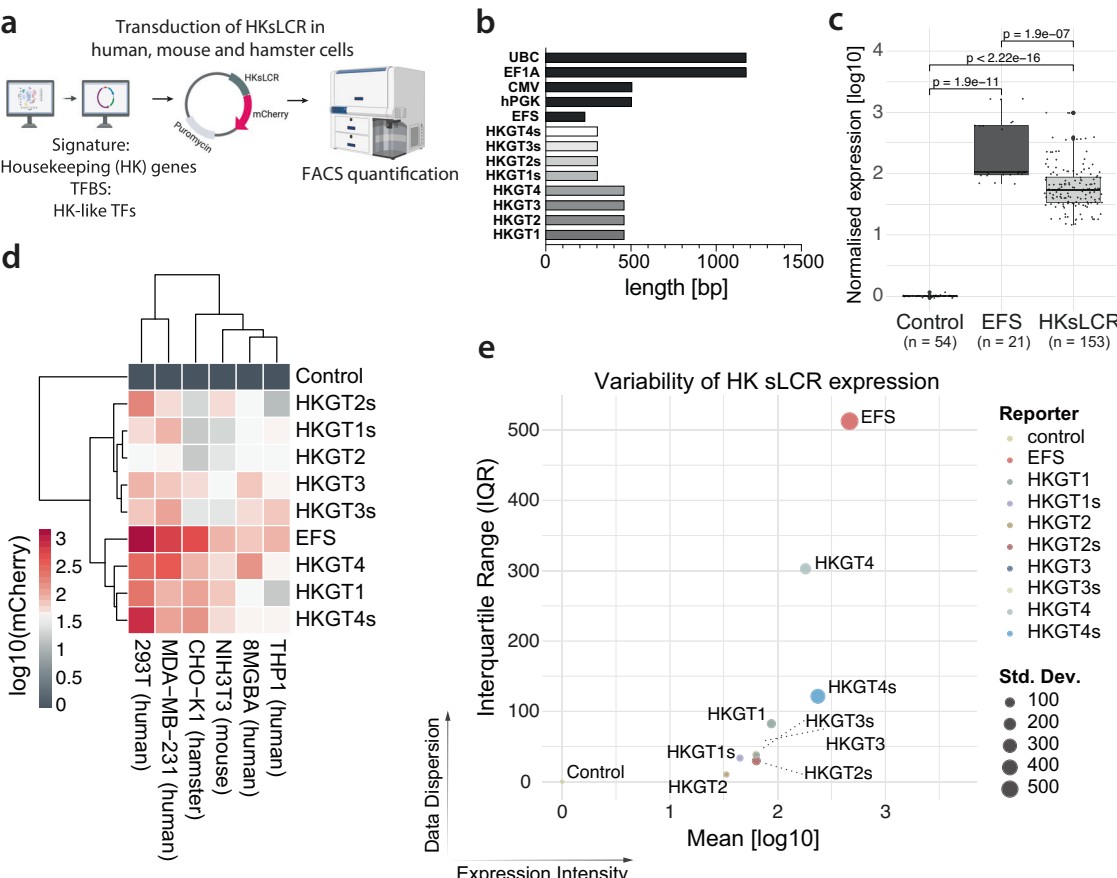

**Fig. 5 | Design and validation of housekeeping-like sLCRs for gene therapy compatible with AAV-vectors size constraints. a** Schematic generation and validation of sLCRs based on broadly expressed genes and transcription factors in humans and rodents (partially assembled using BioRender.com). **b** Bar plot of the sequence lengths of housekeeping-like sLCRs (HKGTs) and well-known broadly expressed promoters. Note all HKGTs (housekeeping genetic tracing sLCRs) to be shorter than well established UBC, EF1A, CMV and hPGK promoters. **c** Box plot of normalized mCherry fluorescence intensities for EFS or HK sLCRs transduced stably in human, mouse, and hamster cell lines. Data distribution is shown, with box indicating the interquartile range and inner line indicating the median. Whiskers extend to represent the data range, including outliers. Small dots denote individual datapoints for triplicate measurements of cell lines of human, mouse, and hamster origin ($n = 6$), genetically engineered with each of the nine constructs. Statistical significance values were calculated by two-sided unpaired $t$-test. **d** Heatmap of the normalized mCherry fluorescence intensities of EFS or HK sLCRs transduced stably into human, mouse, and hamster cell lines. **e** Scatter plot with varying dot size indicating mean log10-transformed intensities and interquartile range of HKs LCR-driven and EFS-driven mCherry. Dot size represents standard deviation, color-coded for corresponding identities. Note that EFS is the strongest promoter, and HKGT4 is the strongest promoter with lowest variability across all the lines tested. Source data are provided in the Source Data file.

54 individual stable cell lines, encompassing human blood, brain, breast and kidney tissues of origin, as well as rodent species (mouse and hamster). Collectively, all HKGTs showed significantly higher expression than the background ($p < 2.22e-16$; Fig. 5c and Supplementary Fig. S5c), confirming the functionality of LSD-designed sLCRs without minimal promoters and extending their use to low-size sLCRs. Of note, when examining quantitative expression patterns through hierarchical clustering, HKGT1, HKGT4, and short-HKGT4 (HKGTs) demonstrated a similar trend to the strong EFS promoter. Among them, HKGT4 exhibited the highest expression levels, surpassing EFS in various metrics that indicate reduced variability in cell-to-cell expression (Fig. 5d, e and Supplementary Dataset S1).

Among the best-performing sLCRs HKGT1, HKGT4 and HKGT4s, we observed enrichment of SP1-like and Kruppel-like motifs, and a RBPJ motif was shared between the strong EF1A/EFS promoters and HKGT4/HKGT4s (Supplementary Fig. S5d). Despite the limited sample size, these factors have strong biochemical evidence supporting their role as drivers of broad and high-volume gene expression[38,39].

To extend our strategy to tissue-specific sLCRs in silico, we curated a list of signature and transcription factor genes associated with the human kidney and ran the LSD pipeline using kidney single-nuclear-ATAC-seq data[40]. To control for specificity, we designed two additional sLCRs with curated inputs for human lung and liver tissues and enriched for CREs incorporating pan-cancer ATAC-seq data (Methods). In silico comparisons using ssGSEA, TFBS correlation, and phenotypic ranking consistently demonstrated the potential of the designed sLCRs to exhibit tissue-specific expression (Supplementary Fig. S6).

Together, the data support the ability of LSD to design AAV-compatible sLCRs for gene therapy.

### LSD enables prioritizing cell-state-specific drivers in combination with genome-wide gain-of-function CRISPR screens
The generation of cell-state-specific reporters enables the discovery of cell-state regulators by hypothesis-driven approaches and also by unbiased genetic screens[14]. To illustrate the experimental opportunities enabled by designing sLCRs to represent multiple cell states, we performed genome-wide pooled CRISPR activation screens (CRISPRa) in Isocitrate dehydrogenase wildtype human glioma-initiating cells (IDH-wildtype-hGICs) with either MGT1, MGT4 or PNGT3 as a readout. This strategy potentially allows the discovery of genes whose

transcriptional amplification leads to a mesenchymal or proneural cell fate commitment.

A CRISPRa library targeting 18,915 human genes via 104,540 on-target and 1895 control sgRNAs (~5 sgRNAs/gene)[41] enabled us to carry out a genome-scale gain-of-function phenotypic screen. The presence of a fluorescent marker in each vector directly supports that the conditions of transduction respected the one guide per cell rule (multiplicity of infection [MOI] <0.5), while maintaining a minimal library representation throughout the cell culture experiment (i.e. ~200x). We FACS isolated the cells with the highest and lowest expression of the reporter for each GBM subtype sLCR after seven passages and fifteen days of gene activation through specific sgRNAs, a time frame designed to permit transcriptional activation and cellular reprogramming (Fig. 6a).

At the experimental endpoint, there was a substantial equilibrium between guide-bearing and non-infected cells and the reporter expression appeared independent of the library (Supplementary Fig. S7a). This suggests that the library overexpression might have introduced mild non-autonomous cell fate changes, but – more importantly – it excluded bleed-through technical artefacts that may complicate the FACS sorting of phenotypically different cells based on fluorescence. Next, we subjected the candidate outliers to FACS sorting, genomic DNA extraction, PCR amplification and sequencing of the sgRNA library, similar to our recent sLCR-driven phenotypic loss-of-function screens[14,42]. Sequencing of the plasmid library confirmed that the qualitative representation was maintained. Quantitatively, each condition clustered independently, indicating that sorting subsets of cells by reporter expression introduced a selective segregation of the sgRNAs (Supplementary Fig. S7b, c). When comparing all cell-based conditions to the library, only a limited subset of sgRNA targets scored as significantly differentially abundant (Supplementary Dataset S2). There was a notable enrichment of cell cycle promoters in cells, including Cyclin D1 and Cyclin D3 (Supplementary Fig. S7d), indicating that active sgRNAs in the library can functionally amplify biological programs (Supplementary Fig. S7e).

Unbiased analysis of the dataset using robust rank aggregation and enrichment over the reporter-low fraction uncovered genes whose potential amplification by multiple sgRNAs could drive reporter-specific upregulation (Fig. 6b and Supplementary Dataset S2). Strikingly, we found RELA among the MGT4 screen top hits (Fig. 6b and Supplementary Dataset S2): RELA is one of the NF-kB transcription factors that we previously identified as an MGT1 regulator under homeostatic conditions in glioblastoma-initiating cells in both focused loss-of-function experiments and genome-scale CRISPR KO screens[14]. Since we previously showed that MGT1 is inducible via multiple signaling pathways under in vitro settings[14], we next focused our analysis on common MGT1 and MGT4 candidate hits as a means to globally illustrate the discovery power of the combined LSD-CRISRPa approach. Ingenuity Pathway Analysis (IPA) revealed that the mTOR (padj = 1.45e−7), BEX (padj = 3.39e−6), WNT (padj = 2.45e−4) and EMT canonical (padj = 6.17e−6) pathways connected genes defined by two sgRNAs with significant positive enrichment in both MGT1 and MGT4 screens (median log2 Fold Change >0). In particular, the EMT pathway gene set was significantly more enriched in MGT1 ($p = 1.7e−5$) and MGT4 screens ($p = 6.3e−5$) when compared to PNGT3 (Fig. 6c). This suggests that reporter-expressing cells correctly captured the gain-of-function activity imparted by CRISPRa sgRNAs, which are likely candidates for a PMT.

To overcome the limitations imposed by each experimental platform, we reasoned that genuine phenotypic drivers could be enriched as MGT1-4 and PNGT3 sLCR drivers, which converge on differential expression in glioblastoma biopsies (TCGA GBM, Verhaak et al.[21]) or pathway activation (Garofano et al.[23]) and glioblastoma stem cell state-specific dependencies identified by systematic CRISPR genome-wide KO screens[22] (Fig. 6a, d–e). This

approach prioritized candidate drivers of a proneural-to-mesenchymal transition (PN-to-MES), including the extensively validated pro-mesenchymal drivers WWTR1[21] and FOSL1[31], also conserved in the proliferative-to-glycolytic (PPR-to-GPM) axis (Fig. 6d, e). Indeed, constraining the analysis to the GBM-subtype-specific dependencies leads to a 156.32 and 171.95 fold enrichment over expectations for overlap with MGT1 and MGT4, respectively ($p = 1.36e−16$ and 5.39e−17, respectively; Supplementary Fig. S7f). Ranking of the top 500 hits by fold changes maintained a significant overlap between MGT1 and MGT4, albeit far less enriched (7.8%, $p = 1.77e−09$). Together, these analyses indicate that convergent hits from these orthogonal experiments effectively restrain the background contributed by the use of different disease models and technical limitations associated with either experimental approach (see Discussion). Importantly, both screens featured the eight shared novel candidate drivers of PMT (Fig. 6e–g and Supplementary Fig. S7f), and each reporter identified specific candidates, including WWTR1[21] by MGT1, and FOSL1[31] by MGT4. Both WWTR1 and FOSL1 are well-established mesenchymal-GBM transcription factors and could have been prioritized based on the intersection of the Verhaak or Garofano and Richards datasets and prior knowledge. Instead, the other hits share features that make them likely but non-obvious novel candidates for a cell-intrinsic PMT driver function. Overall, among the eight shared hits featured by our orthogonal analysis as drivers of a mesenchymal program activation, 37.5% (3 out of 8; KLF5, MED7, ZFP36L1) point to transcription, and 62.5% (5 out of 8; PGK1, RAC1, COBP2, ZNHIT2, SPTLC2) point to regulation of metabolism as a feature of the mesenchymal program.

The intersection of the Verhaak and Richards datasets alone appears to be insufficient to define bona fide proneural regulators beyond obvious ones such as OLIG2 and ASCL1 (Fig. 6d). Importantly, the orthogonal combination of all three approaches defines a subset of 38 hits. Globally, by Ingenuity Pathway Analysis, the PNGT3-high hits featured the activation of the EIF2 pathway ($p = 5.64e−09$) microRNA biogenesis ($p = 7.93e−08$) and Huntington's disease signaling ($p = 5.73e−07$) pathways, and the MYC transcriptional program activation ($p = 5.3e−15$). The high proneural identity displayed by our in vitro models might limit the discovery power of the PNGT3 screen and suggests that mesenchymal GSCs or signaling may provide better ground for screening proneural amplifiers. However, the identification of the adhesion G protein-coupled receptor B1 (ADGBR1), also known as brain-specific angiogenesis inhibitor 1 (BAI1), is consistent with a role in proneural differentiation. Indeed, BAI1 is a synaptic receptor whose overexpression limits neo-vascularization and xenograft formation[43] and correlates with APLRN-driven invasive potential in proneural glioblastoma cells[44].

To validate our screen, we employed four non-overlapping sgRNAs[45], which offer robust and superior gene activation compared to the single sgRNAs used in the genome-wide screen. As candidate PMT drivers, we focused on RELA and RAC1, featuring unbiased and orthogonal hits, respectively, as well as FOSL1, which features both a hit and a control. Over a period of 14 days comparable to the genome-wide screen, all hits markedly promoted MGT4 expression (Supplementary Fig. S7g). RELA and FOSL1 markedly induced MGT4 already at an early timepoint, whereas RAC1 did not (Supplementary Fig. S7g), suggesting that temporal control of gene activation may be critical to defining different classes of drivers. FAC-sorting and RNA-seq of reporter-high-expressing cells revealed that RELA, FOSL1 and RAC1 were specifically and robustly induced by their respective sgRNAs and were all co-regulated along with the expected GBM gene sets (Supplementary Fig. S7h, i). A PPR-to-GPM/MES2 transition was evident in our cellular model, thereby establishing a direct connection between CRISPR activation of candidate drivers, MGT4 expression and a cell identity change at endogenous gene expression level (Supplementary Fig. S7h, i).

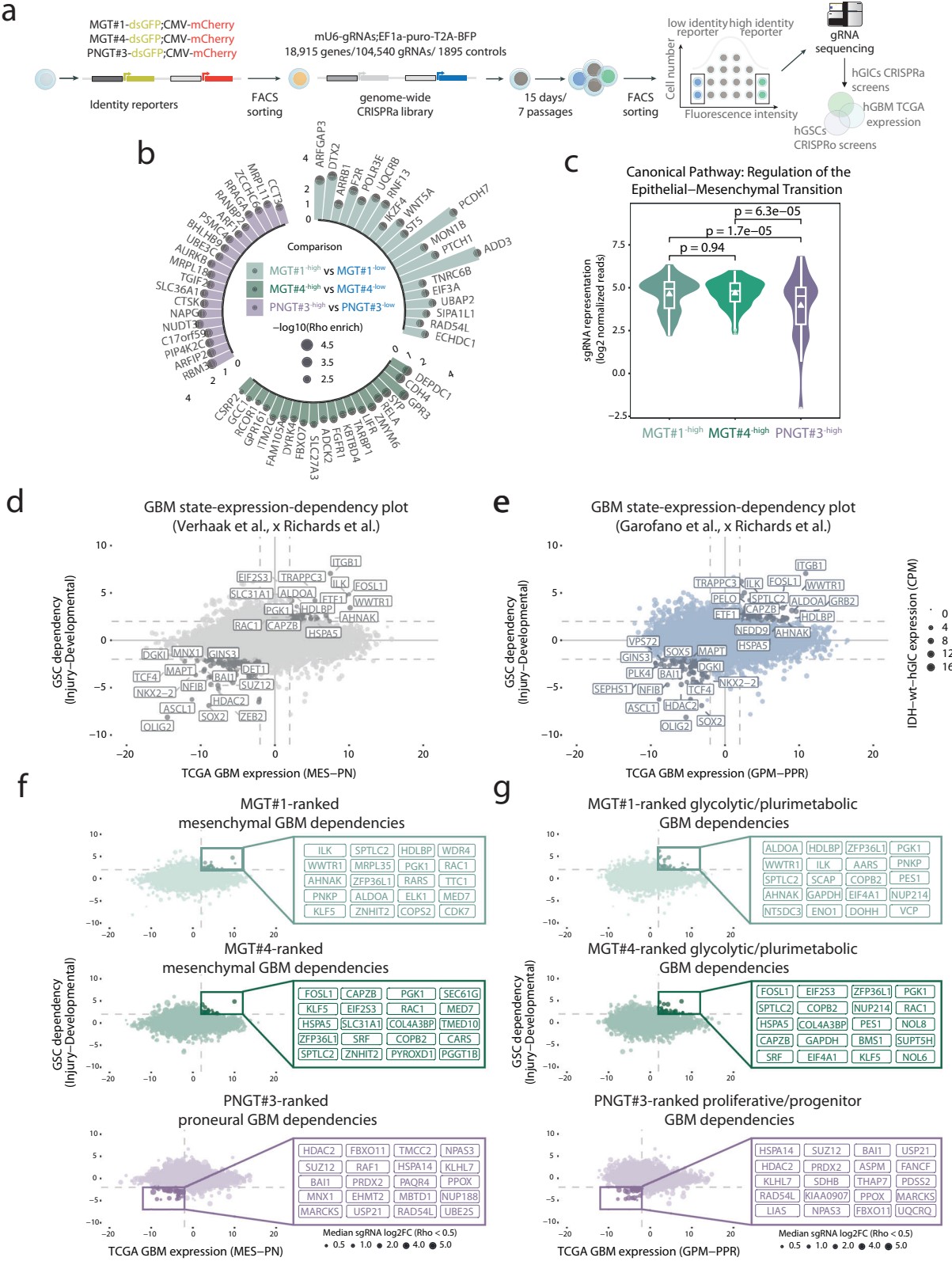

Together, the experimental combination of LSD-designed sLCRs, CRISPR screens and patients' molecular maps uncovered the bona fide regulators of cell states that connect the tumor genotype to its molecular phenotype. This showcases the utility of LSD as a generally applicable framework to study cell identities and cell state changes.

## Discussion

Here, we present LSD as a framework to streamline the tracing of cell identities and state changes for complex phenotypes through synthetic genetics. LSD scans the reference genome for candidate *cis*-regulatory DNA using one list of biomarkers and one list of TFs defined as active in the cell type or state of interest. Using previously validated

**Fig. 6 | Convergence of LSD, genome-wide CRISPR activation and patients' datasets towards the discovery of cell-state-specific drivers. a** Schematic representation of the genome-scale CRISPR activation (CRISPRa) screens (partially assembled using BioRender.com). **b** Circular barplot depiction of the top single-guide RNAs (sgRNA) targets with a positive enrichment (median sgRNAs log2-fold-change (log2FC) >0.5, Rho Enrichment score <0.05 and gene expression in target cells CPM >5) in the indicated high-reporter expressing cells compared to the respective controls (color legend, $n = 3$ technical replica). Median sgRNA log2FC and Rho enrichment scores are represented by bars and dot size, respectively. c) Violin plot of the read-count distribution of sgRNA targets defined as connected to EMT terms by Ingenuity Pathway Analysis ($n = 70$). Data distribution is shown, with box indicating the interquartile range and inner line indicating the median.

Whiskers extend to represent the data range, including outliers. *P*-values denote the significance of statistical comparisons by two-sided pairwise *t*-test. Scatter plot depicting The Cancer Genome Atlas (TCGA) GBM subtype-specific expression according to Verhaak et al.[21] (**d**) or Garofano et al.[23] (**e**) classifications (X-axis) and GBM stem cells dependencies (Y-axis). Dot size represents gene expression values in naive IDH-wildtype hGICs. **f**, **g** Convergence of the indicated LSD-CRISPRa-screens data onto the GBM-state-expression dependencies plots in **d** and **e**. Dot size represents the median log2FC of the respective sgRNA targets. Note that top right/bottom left data highlight candidate GBM subtype-specific dependencies, with the top context-specific factors listed in the magnification. LSD logical synthetic *cis*-regulatory DNA, hGICs human glioma-initiating cells, EMT Epithelial-Mesenchymal-Transition. Source data are provided in the Source Data file.

GBM sLCRs[14] as a benchmark, it is shown that the LSD primary output outperforms previous designs in silico and is functional and specific in head-to-head experimental validation, including a genome-wide CRISPRa screen. Whereas the utility of LSD is straightforward in the largest set of cases for which surface markers for cellular phenotypes are not available, our data suggest that LSD may also be preferable to using CD24 and CD44 as canonical markers for epithelial and mesenchymal states, or in combination. The seamless generation of genetic tracing reporters for distinct cellular phenotypes by LSD represents a resource for future experimental validation (Supplementary Dataset S1). Our modeling approach on validated mesenchymal GBM sLCR shows that both high levels of target TFBS (Qscore) and enrichment of TFBS at signature genes (SignScore) are higher in mesenchymal LSD reporters than in those specific to different phenotypes or previous selection. This underscores the importance of curated TFBS inputs to advance specificity of the reporters, which will likely benefit from the recent development of CUT&RUN/TAG for TF footprinting[46] and tools for TFBS prediction and gene regulatory network inference[47,48]. Broadly, despite LSD's ability to generate a selection of CREs that outperforms random selections, a better understanding of sequence-specific gene regulation and high-quality input are likely critical for effective learning and ranking by LSD. Hence, iterative refinement that combines LSD upstream computational design tools[49] and downstream massively parallel testing[8,12] holds the potential to improve synthetic reporter design even further.

The advent of high-throughput chromatin accessibility profiling enabled mapping gene regulatory landscapes in healthy and diseased cell states but outpaced the development of experimental methods to directly test hypotheses generated from these data. Moreover, bona fide cell states identified by single-cell data ex vivo require in vivo validation, as artefactual signatures may be confounding[50]. Previous methods successfully leveraged *cis*-regulatory DNA and expression datasets to deploy cell-type-specific enhancers and enable genetic tracing and perturbation of gene function across mammalian cell types[9,11]. Endogenous or minimal viral promoters were key to generating functional and specific synthetic promoters in past approaches[11,25]. Similar to our previous design of sLCRs[14], LSD generates functional reporters without minimal promoters, which is a key advantage to increasing phenotype specificity. Moreover, we developed a ranking system to assess the on-target potential of genetic reporters. With this combination, we designed the second generation of glioblastoma sLCRs, as well as other sLCRs targeting a diverse set of molecular phenotypes. To showcase the versatility of LSD in addressing synthetic biology challenges, we employed it to design short functional synthetic promoters, eliminating the need for non-specific minimal promoters, a common issue in AAV vector-based gene therapy. In parallel work, we merged LSD and phenotypic ranking strategies to swiftly generate functional reporters for epithelial response to SARS-CoV-2 during the COVID-19 pandemic. These reporters faithfully replicated pathophysiologically relevant cell states observed

in patients' samples, providing a robust platform for high-throughput screening of potential therapeutics and their mechanisms of action[51].

The incorporation of custom inputs, such as chromatin accessibility and 3D contact maps, offers novel applications for LSD. These range from validation of atlases of cell identities to gene therapy. However, systematic, large-scale and in vivo testing is necessary to fully explore these possibilities. In this study, we addressed a specific aspect of AAV design: the size limitation of promoters for robustly expressing large cargoes. AAV-based clinical trials, especially in retinal diseases, highlight the timely relevance of this approach. We applied LSD to successfully design and validate eight HK sLCRs in stable cell lines from human and rodent tissues. The process involved utilizing input lists of highly expressed housekeeping signature and TF genes, along with chromatin accessibility data. Notably, HKGT4, designed based on a conserved human-mouse signature, exhibited high expression levels and reduced cell-to-cell variability compared to the EFS promoter. These results emphasize LSD's potential to broaden AAV-compatible sLCRs for gene therapy. However, achieving optimal transgene expression and cell-type specificity remains a key challenge in vector design, requiring precise delivery to target cells while avoiding unintended effects in non-target tissues. The use of HK promoters is sufficient in cases where targeted delivery is attainable, while in all other cases, tissue-specific *cis*-regulatory elements will be necessary, tissue-specific *cis*-regulatory elements will be necessary. We demonstrated that designing tissue-specific sLCRs provides specificity in silico, but in vivo translation will necessitate specific tissue focus, deep gene expression, and chromatin accessibility datasets, along with attention to in vivo delivery. LSD's systematic success in designing functional promoters suggests its potential as a blueprint for developing tissue-specific promoters, further enhanced by cell-type-specific TFBS, which has proven effective in focusing on specific transcriptional programs[52–55] and may be now automated using deep learning[56,57].

We previously showed that sLCRs connect changes in the mesenchymal state of glioblastoma or lung cancer cells to the loss of function caused by CRISPRout or CRISPRi in pooled screens[14,42]. Here, we expanded the use of sLCRs in phenotypic pooled screens to gain-of-function by using CRISPRa. Unlike the proneural sLCR, the mesenchymal sLCRs promoted the representation of gRNAs targeting EMT genes among dCas9-VPR and gRNA-bearing cells. In combination with patients' gene expression profiles[21] and functional screens in patient-derived cells[22], genome-scale CRISPR activation in sLCR-containing cells pointed to novel candidate drivers of individual glioblastoma transcriptional phenotypes, including genes not previously described as drivers. This illustrates the utility of combining descriptive and functional information to prioritize candidate targets and biomarkers for validation. Of note, this approach may be particularly helpful in understanding the molecular foundation of glioblastoma subtypes and states whose significance is still unclear. In line with the reclassification of glioblastoma states by pathway activation[23], our analysis uncovered several metabolic regulators. Albeit we cannot exclude that metabolic

regulators may be promoted owing to fitness advantage rather than a bona fide pro-glycolytic and pro-mesenchymal role, the partial convergence of the distinct mesenchymal and glycolytic GBM was previously noted by Garofano et al.[23]. For example, PGK1 is a well-established HIF1a-regulated gene, and its upregulation is more obvious in differentiated glioma cells under hypoxia as compared with matched GSCs[58], which is consistent with both a pro-fitness and pro-glycolytic/mesenchymal activity. Likewise, the zinc-finger protein ZNHIT2, albeit poorly characterized, is involved in both the spliceosome machinery and potentially connected to mTOR-regulated response to energy- and nutrient-sensing stress[59]. Collectively, gRNAs enriched in the screen were particularly enriched for the mTOR pathway, connecting individual drivers to global metabolic adaptation and PMT. Notably, the identification of known mesenchymal GBM drivers such as WWTR1[60] and FOSL1[31] by MGT1 and MGT4, respectively, denotes the specificity of the sLCRs and suggests that RAC1 could act as a driver of PMT, albeit with lower potency or cooperatively. Consistently, RAC1 was previously connected to mTOR-dependent growth control and STAT3 signaling in NF1-deficient cells[61], the main genetic driver of mesenchymal GBM[21]. Future improvements should encompass more sophisticated culture conditions for GBM cells in vitro that take into account heterogeneous identities and reflect specific signaling and metabolic requirements. The constant development of powerful dCas9 effectors and arrayed gRNA libraries is likely to improve the discovery power of genome-wide phenotypic screens[45]. Arrayed screens are also anticipated to reduce the influence of non-autonomous phenotypic changes in pooled screens, which are responsible for neutral gRNA background recovery in multicellular 3D models like the one shown here.

Applying Boolean logic gate strategies[62], including regulators of mRNA translation and stability as well as protein homeostasis regulation, will further enhance accuracy and specificity of complex phenotypes' synthetic genetic tracing.

In conclusion, we expect LSD to provide a simple and scalable approach to performing genetic tracing in complex multifactorial settings, enable validation experiments of omics approaches on equal complexity terms, as well as to study fundamental questions underlying transcriptional regulation.

## Methods

### Datasets
TCGA data were from dbGaP database of Genotypes and Phenotypes (dbGaP) accession phs000178. The TCGA-GBM gene sets are obtained from Verhaak et al.[21] and Garofano et al.[23] and the GSCs dependencies are from the supplementary material in Richards et al.[22]. Pan-GBM single-cell data from Ruiz-Moreno et al.[24] was downloaded from Zenodo (DOI: 10.5281/zenodo.6962901). CRISPRa screen data generated in this study are attached in Supplementary Dataset S2. Tabula Muris database was accessed through https://tabula-muris.ds.czbiohub.org/ and data was downloaded from Figshare. Tabula Sapiens database was accessed through https://tabula-sapiens-portal.ds.czbiohub.org/ and data was downloaded from figshare. Gene sets for Gene Set Enrichment analysis and for pre-filtering of sLCR input were downloaded from Molecular Signatures Database (MSigDB, https://www.gsea-msigdb.org/gsea/msigdb). To define tissue-specific TF genes, we utilized data from the Protein Atlas (www.proteinatlas.org).

### LSD algorithm
The LSD algorithm takes a list of position weight matrices (PWMs) along with the corresponding cognate TF-genes, a list of marker genes of a target phenotype, and the reference genome of the organism of interest, and it generates a list of naturally occurring, putative cis-regulatory elements, that are then used to assemble the synthetic reporter for the target phenotype. The algorithm can be divided into three steps. In step I, LSD generates a pool of potential CREs with a fixed length within user-defined regulatory landscapes (default is a 150 bp window sliding with a 50 bp step). In step II, LSD assigns TF-binding sites to the CRE pool using FIMO (default -output-pthresh 1e-4 -no-qvalue), and creates a matrix of putative CREs x TFBS. In step III, LSD ranks and selects the minimal number of CREs representing the complete set of TFBS. For that purpose, it iteratively sorts and selects the best CRE based on the overall TFBS affinity and diversity among the input TFs showing high affinity for the CRE. Starting from the ranked CREs, LSD selects the highest-ranking CRE defined by the sum of the affinity score (-log10(p-value)) and TFBS diversity (number of distinct TFs represented in the predicted TFBS). Subsequently, it removes the selected CRE and the corresponding TFBS from the CRE x TFBS matrix and repeats the selection. This continues until either none of the CRE or of the TFBS is left. In the ranking, priority is given to CREs proximal to known TSS to increase the chances of successful transcriptional firing using the same strategy as above. TSS were defined based on FANTOM CAGE peak BED files (RRID:SCR_002678; Human: http://fantom.gsc.riken.jp/5/ datafiles/phase1.3/extra/TSS_classifier/TSS_human.bed.gz and TSS-containing CREs were defined based on overlap with genome-wide pooled CRISPRi libraries[63]. Finally, LSD returns an ordered list of the selected CREs, together with a representation of the TFBS scores (Fig. 1a). The framework to run the LSD algorithm is available at: https://gitlab.com/gargiulo_lab/sLCR_selection_framework.

### sLCR design by LSD
Reporters designed using the LSD method are indicated in Supplementary Dataset S1. First-generation GBM-sLCRs[14] were designed by manual integration of a selection of top-ranked CRE. LSD GBM-sLCRs (Fig. 1) used as input the first generation sLCR gene signatures and a selection of subtype-specific TFs (high-expressed TF genes, >quantile 75%; https://meme-suite.org/meme/) obtained from TCGA-GBM patients' RNA-seq expression profiles (RPKM-UQ, TCGA, phs000178.v3.p3). Random Sign/TF sLCRs were generated by integrating different sizes of randomly selected human genes (three different samplings of 10, 20, 50 and 100 different genes) and TF genes (four samplings of 1 to 301 different TF). Random TF only used MGT1-4 signature genes to generate the CRE pool. AC-like state LSD-sLCRs were generated using Neftel et al. AC-like signatures[19] and CLGT1-2-3 TFBS. Retinal LSD reporters were generated using retina-specific cell-population marker genes (GSE81905) TF genes determined at different thresholds (>quantile 60–88%). hg19 (GBM) and mm10 (retina) assemblies were used to extract genomic positions. Reporters designed using the LSD method are indicated in Supplementary Dataset S1. First-generation GBM-sLCRs[14] were designed by manual integration of a selection of top-ranked CRE. LSD GBM-sLCRs (Fig. 1) used as input the first generation sLCR gene signatures and a selection of subtype-specific TFs (high-expressed TF genes, >quantile 75%; https://meme-suite.org/meme/) obtained from TCGA-GBM patients' RNA-seq expression profiles (RPKM-UQ, TCGA, phs000178.v3.p3). Random Sign/TF sLCRs were generated by integrating different sizes of randomly selected human genes (three different samplings of 10, 20, 50 and 100 different genes) and TF genes (four samplings of 1 to 301 different TF). Random TF only used MGT1-4 signature genes to generate the CRE pool. AC-like state LSD-sLCRs were generated using Neftel et al. AC-like signatures[19] and CLGT1-2-3 TFBS. Retinal LSD reporters were generated using retina-specific cell-population marker genes (GSE81905) TF genes determined at different thresholds (>quantile 60–88%). hg19 (GBM) and mm10 (retina) assemblies were used to extract genomic positions.

Multiple sequence alignments were conducted with four different tools (MUSCLE, Clustal Omega, T-Coffee and MAFFT in SnapGene) with similar outcomes. MUSCLE output is reported in Fig. 2a.

## Housekeeping-like sLCR design by LSD

To identify housekeeping genes (HKG), we utilized two datasets: Tabula Sapiens[34] and Tabula Muris[35]. First, for Tabula Sapiens, we filtered out genes and cells with low counts (<q25%) and (<q5%), respectively, to eliminate potential outliers and select HKG candidates. We then retained only genes expressed in at least 75% of the cells, resulting in a list of 392 HKG gene candidates. We further refined this list by removing genes below the >25% total counts filter, as well as mitochondrial and ribosomal genes. This refinement yielded 206 robust HKG candidates in Tabula Sapiens. For Tabula Muris, we followed a similar approach, filtering low-count genes and cells. In total, we identified 123 HKG candidates. To rank the genes based on cellular expression, we selected the top 30 HKG from each dataset to define the final list.

Next, to identify potential HKG transcription factors (TFs) within the context of Tabula Sapiens and Tabula Muris datasets, we correlated the TFs expressed in the above lists with those from various TF databases, including our compiled TF selection. This analysis led to the identification of 9 TFs in Tabula Sapiens and 4 TFs in Tabula Muris. To enhance the potential of the reporters, we included CTCF, RAD21, and YY1 as potential structural elements contributing to expression stabilization. To assess TF binding potential, we examined TF binding on the sequences of strong promoters such as EFS, EF1A, UBC, hPGK, and mPGK. This analysis identified 381 TF motifs with varying degrees of interaction. We further evaluated the overlap of TF datasets, enriched TFs on strong promoters, and compared them with Joshi et al., HOUNKPE_HOUSEKEEPING_GENES.v2022.1.Hs.gmt databases, and our own selection. In order to increase the number of reporters, we also employed a general selection of all reporters shared and present in the EFS. Overall, we obtained a non-redundant set comprising 29 core TFs and 84 general TFs for further analysis in the generation of the reporters.

On the final HK signature and TFs lists (Supplementary Dataset S1), we employed LSD to generate the reporters. We chose the top 2 cis-regulatory elements (CREs) to generate HKGT1s-4s and added one extra CRE for HKGT1-4, each consisting of 300 base pairs, which closely resembled the strong promoters. In the process of generating these reporters, we selected only CREs that were present in the pan-cancer TCGA ATAC-seq data for human and scATAC-seq data for mouse. This approach ensured that the reporters captured the desired characteristics and were derived from active genomic regions.

## Tissue-specific sLCR design by LSD

To generate a signature selection of tissue-specific LSD-sLCR, we implemented a filtering process on the cell populations of the kidney, liver, and lung using the Tabula Sapiens dataset. Our approach involved evaluating markers across tissues based on the following criteria: p_val_adj <0.05, avg_log2FC>1, and pct.2>0.1. Subsequently, we identified tissue-specific populations that were annotated for each tissue in the protein Atlas expression data. Additionally, we conducted manual curation specifically for the kidney list, aiming to identify representative kidney signature genes. This curation process resulted in a total of 25 signature genes for the kidney, 117 for the liver, and 10 for the lung. Next, we proceeded to identify tissue-specific TF (transcription factor) genes by cross-referencing the list of signature genes with our comprehensive TF database. To define tissue-specific TF genes, we utilized information from the Protein Atlas (www.proteinatlas.org), incorporating both criteria. For the kidney dataset, we also employed chromVar[64] results from the publication found at https://www.nature.com/articles/s41467-021-22368-w#Sec34 (The dataset can be obtained from this reference's Supplementary Data 3), in conjunction with the aforementioned filters. This additional analysis yielded an enhanced kidney list comprising 40 TF genes, 16 TF genes for the liver, and 15 TF genes for the lung. Finally, we employed the LSD-sLCR pipeline to generate the reporters, which we subsequently validated using the aforementioned method.

## Phenotypic potential inference by on-target TFBS scoring

The inference of the specific phenotypic potential of each LSD-reporter was defined as the linear correlation between the Qscore (affinity-score) and the SignScore (phenotypic-score). The affinity-score (QScore) was calculated for each reporter as the sum of the TFBS-affinity using FIMO (default -output-pthresh 1e-4 -no-qvalue) given a specific set of TFBS (e.g., MGT1-2 TFBS) and normalized by the sLCR sequence length and the ratio of observed/expected TFBS ($\sum$ (FIMO-score)/sequences length) * (observed/expected TFBS))). In contrast, the phenotypic score (SignScore) was defined as the ssGSEA-enrichment value calculated on target expression profiles (ssgsea.norm = FALSE) using the signature genes, and normalized by the ratio of observed/expected TFBS (ssGSEA-score * (ratio observed/expected TFBS)). Scatter plots were generated using ggplot2 on an R v.3.6 environment.

## Evaluation of the CRE selection

To model the number of CRE required to account for 50% of the total TFBS potential, we used experimentally validated LSD vectors and fitted a lasso regression model. To begin, we generate a list of all the top-CRE sequences for each LSD-sLCR generated. Then, we used FIMO (default −output-pthresh 1e-4 −no-qvalue) to map their corresponding TFBS to each sequence and calculated the ratio of observed/expected TFBS for each top-CRE selection. Finally, we use the lasso function to fit the top-CRE and TFBS ratios. The lasso function was used for the fitting and the ggplot2 package to generate plots (R v.3.6).

## Comparison of LSD reporters

The analysis of differences between GBM-subtypes enrichment scores was generated by comparison of ssGSEA values (GSVA v.1.32, ssgsea.norm = TRUE, and default parameters) for each synthetic reporter. To evaluate the similarities between GBM-sLCR transcriptional potential, we evaluate the correlation between TFBS-affinity of all GBM-TFBS and associated TF/TFBS for each GBM-sLCR. The evaluation of the signal-to-noise ratio in ME-GBM was calculated by defining the MGT1-2 observed/expected TFBS ratio FIMO (default -output-pthresh 1e-4 -no-qvalue) using GBM-sLCR and random sLCR. The evaluation of the correlation between the phenotypic potential in GBM-ME and Amacrine cell was generated by computing the phenotypic-potential correlation using MGT1-2 and Amacrine cell TFBS (exclusive high-expressed TF > 75%; below) on each synthetic reporter. The signature enrichment was calculated by using the average of ME-GBM patients' expression profiles and Amacrine cells single-cell expression profiles. PromoterDB sequences were retrieved from publication (https://data.fmi.ch/promoterDB/) and integrated into the phenotypic-potential analysis with other design LSD-sLCRs. The inference of the phenotypic potential of LSD ACL reporters was generated by using CLGT1-2 TFBS and the average of CL-GBM patients' expression profiles. Graphical representations were generated using ggplot2 in an R v.3.6 environment.

## Integration of random sLCRs designed by LSD

To evaluate the signal-to-noise ratio for the LSD-generated reporters, we generated LSD reporters using randomly selected TF and signature genes. First, we sampled using the sample R function different sizes of TF (from 1 to 301 by 20 each, without replacement) from the same input TF database used to generate the LSD and sLCR reporters (repository). Then, in the same way, we sampled different sizes of signature genes (10, 20, 50, and 100 genes) using the same assembly to generate the sLCR reporters (hg19).

To generate the reporters, we used the LSD pipeline with default parameters for different combinations of inputs (e.g., random TF and

random signature genes). To evaluate the combination, we evaluated the TFBS on the included LSD-sLCR and first-generation sLCR. The model of the distribution was generated using geom_smooth (default parameters). Finally, the graphs were created with ggplot2 in R v.3.6 environment.

## Evaluation of the Glioblastoma states

To evaluate the distribution of ssGSEA score on Glioblastoma single-cell expression profiles, we used IDHwtGBM.processed.SS2.logTPM.txt from Neftel et al.[19]. The evaluation of the score integrates signature and TF gene expression in those cells. We maintained the separation between AC/MES and OPC/NPC to differentiate between states. The ssGSEA score was generated using the GSVA R package. The graphs were created with ggplot2 in an R v.3.6 environment.

## Bootstrap analysis of LSD reporters

We used a bootstrap approach to rank the differences between LSD reporters generated with an identical set of signature genes and TF but with distinct boundary regions and known CRE filter conditions (Results, Fig. 4a). To begin, we generated several LSD reporters (Methods) using the MGT4 signature genes and CTCF or TAD-defined boundary regions[30]. The LSD approach allows for filtering regions of interest by determining the overlap between those regions and the CRE. We used this functionality to filter CRE obtained from the LSD pipeline using pan-Cancer ATAC-seq regions[29]. Input genes and TF from MGT4 were correlated to the mouse and used as input to generate the mouse LSD reporters. The regulatory regions for mouse LSD were defined from scATAC-seq. After generating the reporters using MGT4 input, we used CRE to conform the LSD reporters. We evaluate the sequences by mapping MGT4 TFBS to each sequence (FIMO; default −output-pthresh 1e-4 −no-qvalue). We used the affinity values for the bootstrap analysis. To compare reporters, we conducted a bootstrap analysis. First, the algorithm generated a reporter-specific distribution of TFBS scores by randomly sampling ($n = 100$) TFBS binding to the reporters (25% of MGT4 TFBS, 60/231). Second, it compares the difference between reporter-specific distributions using the Wilcox test (alternative = greater). Finally, it ranks the reporters according to the number of significant events obtained (adj. $p$-value < 0.05).

## Human Glioma-initiating cell line (hGICs)

The IDH wildtype-hGICs were generated by our lab and will be described elsewhere. Extensive histopathological, molecular and tumorigenic potential characterization has been performed and examples are shared with Editors and Reviewers. Briefly, IDH WT-hGICs were generated by transforming human NPC[65] (kindly provided by R. Glass, LMU), by means of: pRSPURO-sh-PTEN(#1; kind gift from D. Peeper), pLKO.1-sh-TP53 (TRCN0000003754) and pRS-shNF1. For this line, thorough genetic, transcriptional, and epigenetic characterization has been performed, as well as in vivo tumor formation and phenotypic mimicking ability. In vitro, hGICs were propagated as described[66,67] with one modification. In addition to EGF (20 ng/ml; R&D), bFGF (20 ng/ml; R&D), Heparin (1 μg/ml; Sigma) and 1% Penicillin and Streptomycin, PDGF-aa (20 ng/ml; R&D) is also supplemented to RHB-A (Takara). This medium composition will be referred to as RHB-A complete. hGICs were cultured at 37 °C in a 5% $CO_2$, 3% $O_2$ and 95% humidity incubator.

## Transfection/Transduction

Transfection and transduction of Lentiviral constructs were previously described in detail[67]. Briefly, 12 μg of DNA mix (lentivector FH1-MGT1-mVenus_PGK-H2B-CFP, pCMV-G, pRSV-REV, pMDLG/pRRE) was incubated with the FuGENE-DMEM/F12 mix for 15 min at RT, added to the antibiotic-free medium covering the 293T cells, and the first-tap of viral supernatant was collected at 40 h after transfection. Titer was

assessed using Lenti-X p24 Rapid Titer Kit (Takara) according to the manufacturer's instructions. We applied viral particles to target cells in the appropriate complete medium supplemented with 2.5 μg/ml protamine sulfate. After 12–14 h of incubation with the viral supernatant, the medium was refreshed with the appropriate complete medium.

For PiggyBac-vector delivery, AMAXA™ 4D-Nucleofector™ (Lonza, Cologne GmbH) was used for the nucleofection after optimization of nucleofection conditions (Nucleofector® programs and solutions) for the specific cell line. For each transfection reaction, 1.5 μg of pPB[Exp]-mCherry-{MGT1}>d2EGFP or pPB[Exp]-mCherry-{MGT4}>d2EGFP, 0.5 μg of Super PiggyBac Transposase (System Biosciences, PB210PA-1-SBI) and 0.5 μg of piRFP670-N1 (addgene #45457) were used and a final mastermix of DNA and B1.1 buffer with supplements (6 mM KCl, 15 mM MgCl2, 120 mM Na/H2/PO4 pH 7.2 + 50 mM Mannitol added freshly) was prepared. 25 μl of cell suspension mix with DNA was added into each well of the Nucleofection strip. Pulse programme DN-100 was used to deliver nucleofection pulse to each well. Control wells did receive a mock pulse without delivery of actual voltage.

## Cell culture of human and rodent lines

Human HEK293T cells were cultured in DMEM-F12 + 10% FBS + 1% Penicillin/Streptomycin. Murine NIH3T3 cells were cultured in DMEM + 10% FBS + 1% Penicillin/Streptomycin. The hamster ovary-derived cell line CHO-K1 was cultured in DMEM-F12 + 10% FBS + 1% Penicillin/Streptomycin. All lines were cultured at 37 °C and 5% $CO_2$ in a humidified incubator and regularly checked for mycoplasma contamination. Both The MDA-MB-231 and human monocytic THP-1 cells were cultured in Roswell Park Memorial Institute media (RPMI 1640, Thermofisher, 21875091) supplemented with 10% fetal bovine serum (Gibco, 10270106) and 1% Penicillin/Streptomycin at 37 °C in a 5% $CO_2$−95% air incubator. The THP-1 were propagated in the same medium supplemented with 1 mM pyruvate (Life Technologies), and 2 mM GlutaMAX (Thermofisher, 35050-038). Human glioma cell line 8MGBA was cultured in DMEM-F12 + 10% FBS + 1% Penicillin/Streptomycin. All lines were cultured at 37 °C and 5% $CO_2$ in a humidified incubator, were thawed from frozen batches and propagated for a limited number of passages (10-15x), and were screened on a regular basis for contamination using the Mycoplasma Detection kit (Jena Bioscience 11828383, PP-401L).

## sLCR activity transfection screening

Three cell lines of variable species background were plated at a density of 3000 cells (293T and CHO-K1) or 5000 cells (NIH3T3) in their respective medium in black-walled 96-well plates for optical imaging (Greiner, #655090). For transfection on the following day, we used the Fugene HD reagent and determined optimized conditions according to the manufacturers' protocol for each cell line in a 96-well-plate format in a pre-experiment. In brief, we found 100 ng of DNA and varying ratios of Fugene:DNA ratios (293T 2.5:1; NIH3T3 4:1; CHO-K1 4:1) to yield sufficient transfection efficiencies in a total reaction volume of 100 μl per well. Mastermixes from 28 sLCR and three transfection control plasmids with Fugene reagent and serum-free RPMI were prepared accordingly and transferred to the screening plates in triplicate. After 48 h of incubation time, nuclei were stained for 4 h with 2 μM Hoechst 33258 and fluorescent live-cell imaging for Hoechst, GFP, mCherry and iRFP was conducted on a high-content imaging platform (Operetta CLS, Perkin Elmer). We used the non-confocal mode and a 10x air objective to image the whole field of view of each well under live-cell imaging mode with temperature and $CO_2$ control. LED power and detector exposure time were adjusted on non-sLCR transfection controls and untransfected wells.

## High-content screening analysis for sLCR activities

After filtering each fluorescent channel (sliding parabola 10 px), we used the Harmony-Software building blocks to identify and count nuclei

based on Hoechst staining. Fluorescent cell objects were identified based on GFP, mCherry or iRFP intensities. Fluorescent objects were filtered by applying a threshold for object size and mean intensities as well as number of objects were determined. Data with all relevant parameters was exported as .csv files and analyzed using R Studio. To account for differences in transfection efficiencies and to allow cross-comparison of sLCR expression among the three cell lines, we first calculated a transfection score as a proxy for efficiency. From the three non-sLCR transfection control plasmids (pMAX-GFP, UBC-mCherry, piRFP670) we established this score separately per cell line by determining the transfection_score = (control_fluorescent_objects_number/ nuclei_number) for each control plasmid and calculated the combined mean from pMAX-GFP, UBC-mCherry and piRFP670. The value of this score represents the highest fluorescence intensity in the screen for each line and allows scaling of the sLCR plasmid activities between a value of 0 for untransfected controls and 1, as outlined in the following sentence. To assess the sLCR activity in each line, we calculated the sLCR_activity_score = (sLCR_fluorescent_objects_number / nuclei_number) and normalized this value by dividing through the previously established transfection score, which is setting the upper bar for the highest rate of fluorescent activity in each line and allows comparing sLCR activities across cell lines. Mean values of activity scores for each of the 28 sLCRs were calculated from the biological triplicates and data was plotted as boxplots using the ggplot2 package. Statistical testing was done through two-way ANOVA with multiple comparisons testing and Dunnett contrasts p-value adjustment.

## Flow cytometry analysis

For readout of MGT1 or MGT4 upon MES-GBM specific activators, transduced or transfected and sorted hGICs harboring either Lentivirus-MGT1 or PiggyBac-MGT1/4 were seeded as single cells into 6-well plates in RHB-A medium supplemented with 10 ng/ml TNFa (R&D Systems, 210-TA-020) or without TNFa as control. After 48 h culture, hGICs were harvested into single cell suspensions, resuspended into cold RHB-A complete medium and filtered into FACS tubes. For validation of sLCR upregulation in response to dCas9-VPR and sgRNAs from CRISPRa screen hits, cells were maintained for 7–10 passages before harvesting into single-cell suspensions and filtering into FACS tubes. Events were first gated on the basis of shape and granularity (FSC-A vs. SSC-A) and doublets were excluded (FSC-A vs. FSC-H), then appropriate laser-filter combinations to analyze sLCR expression of mVenus, EGFP or mCherry were chosen using a BD LSR Fortessa Flow Cytometer.

For evaluation of housekeeping sLCR expression, human and rodent cell lines were transduced with each reporter and selected with individual concentrations of Neomycin for up to 5–7 passages. Reporter cell lines and untransduced controls were used for FACS readout with analogous pre-gating strategy as above and recording mCherry intensities as final parameter from biological triplicates. Geometric mCherry intensity means for transduced and untransduced cell lines were extracted. Intensity values for each reporter cell line were normalized by division through the mean of untransduced lines for each measurement and log10 transformed. Data variability statistics were calculated on log10-transformed and normalized mCherry intensity values using the sd(), var() and IQR() functions from R. For heatmap, bubble plot, bar plot and ridge plot visualization, ggplot2 (v.3.4.1) was used. Box-plot quantification was done using ggplot2 and pairwise comparisons with Wilcoxon-Rank-Sum test were computed using the stat_compare_means() function from ggpubr (v.0.6.0). FACS data was analyzed and visualized using FlowJo_v10.

For cell sorting, transduced/transfected hGICs were harvested into single cell suspensions, resuspended into cold RHB-A complete medium and filtered into FACS tubes. Sorting was conducted using BD FACS Aria III or Fusion. The appropriate laser-filter combinations were chosen depending on the fluorophores being sorted for. Typically, to remove dead cells, events were first gated on the basis of shape and granularity (FSC-A vs. SSC-A) and doublets were excluded (FSC-A vs. FSC-H). Positive gates were established on PGK-driven and constitutively expressed H2B-CFP as sorting reporter (in case of Lentiviral construct) or iRFP expression (for PiggyBac-transfected cells).

## Human genome-scale CRISPRa library amplification, infection and library construction

For the genome-wide pooled CRISPR activation screen, we utilized the Human Genome-wide CRISPRa-v2 Library (Addgene Pooled Libraries #83978) consisting of 104,540 sgRNAs targeting 18,915 genes (~top 5 sgRNAs per gene). Amplification of the library was performed following the reported Addgene protocol. Viral production was conducted in 293 T cells, with viral titration performed before target cell transduction. To achieve a library representation over 240×, we transduced a total of $5 \times 10e7$ of human glioma initiating cells harboring the dCas9-VPR system as well as mesenchymal or proneural GBM reporters (IDH-wildtype-hGICs MGT1, IDH-wildtype-hGICs-MGT4, IDH-wildtype-hGICs-PNGT3) at a multiplicity of infection of ~0.5. The cells were kept for 15 days, corresponding to 7 passages, after which the cells were FACS sorted for high and low reporter signals. The genomic DNA was extracted by lysing the cell pellets for 10 min at 56 °C in AL buffer (Qiagen, 19075), supplemented with Proteinase K (Invitrogen, AM2548) and RNAse A (Thermo Scientific, 10753721), subsequently purified with AMPure XP beads (Beckman Coulter, A63881), and eluted in EB buffer (Qiagen, 19086). Next-generation sequencing (NGS) libraries were constructed in a two-step PCR setup, where the PCR1 is used to amplify the sgRNA scaffold, while the PCR2 introduced Illumina-compatible adapters with unique P5/P7 barcodes, allowing sample multiplexity. For PCR1, each gDNA sample was divided over 20 to 26 reactions, which were subsequently pooled together and purified using AMPure XP beads. The optimal cycle numbers for PCR2 were determined for 1 μL of each PCR1 individually by conducting a qPCR amplification using KAPA HiFi HotStart Ready Mix (Roche, 7958927001) and 1× EvaGreen (Biotium, 31000). 5 ng of the purified PCR1 of each sample was used as input for the final PCR2. Both PCR1 and PCR2 were performed using KAPA HiFi HotStart Ready Mix. Primers are available upon request. Quality control of the final libraries was performed using the Qubit dsDNA HS kit (Invitrogen, Q32854) and Collibri™ Library Quantification Kit (Invitrogen, A38524500) for quantification and TapeStation High Sensitivity D1000 ScreenTapes (Agilent, 5067-5584) for determination of PCR fragment size. The barcoded libraries were pooled together in equal molarities and sequenced alongside a 10% of PhiX spike on an Illumina Next-Seq500, using the 150 cycles V2.5 chemistry (in a 1 × 150 bp + 8 bp + 8 bp dual-index single-read mode).

## Human genome-scale CRISPRa data analysis

The 20 bp sgRNA sequences were extracted from the reads using cutadapt (v2.1), aligned to the human genome-scale CRISPRa-v2 Library reference using bwa (v.0.7.17-r1188) allowing n 0-2 mismatches to subsequently generate the sgRNA read counts. Undetected sgRNAs and sgRNAs with less than two counts in the reference plasmid libraries were omitted from the downstream analysis. For assessing CRISPRa hits, relevant for the utilized model system, the sgRNA counts were additionally subsetted to targets denoted by high expression in the IDH-wildtype human glioma initiating cells (IDH-wildtype-hGICs) (CPM > 5). Differential sgRNA abundance analysis contrasting high and low reporter-expressing cells was conducted using the gCrisprTools (10.18129/B9.bioc.gCrisprTools; v2.0.0) R package (scoring = " pvalue"). Signals from the non-targeting sgRNA controls were excluded from RRAa aggregation. The top differentially abundant targets (Rho enrichment score <0.05, median sgRNA log2-fold change >1) are visualized in Fig. 6b. Ingenuity Pathway Analysis used as input gCrisprTools-derived fold-changes and FDR values and ranked

canonical pathways and upstream regulators by Fisher's exact test. To identify and prioritize GBM cell state-specific CRISPRa hits, we further performed orthogonal convergence of TCGA differential expression (TCGA GBM, Verhaak et al.[21], Garofano et al.[23]) together with the glioblastoma stem cell state-specific dependencies (Richards et al.[22]). For follow up and subsequent validation, we prioritized CRISPRa hits, that characterized by high TCGA GBM patients cell state-specific expression (Significance Analysis of Microarrays (SAM) expression score > | 2|), enrichment in glioblastoma stem cells CRISPRout genome-wide screening (z-scores >|2|) and median sgRNA log2FC > 0 and Rho enrichment score <0.5 in our LSD-sLCR CRISPRa screens.

## TCGA data analysis

The Cancer Genome Atlas (TCGA) IDH-wildtype GBM patients data profiled with the Agilent 244 K Custom Gene Expression array (G4502A_07_1/2) were obtained using the TCGAbiolinks R package (v.2.22.4[68]) as log2 lowess normalized sample/reference signal (cy5/cy3) ratios collapsed by gene symbol. The missing microarray expression data were imputed using nearest neighbor averaging via the impute R package (v.1.68.0). For the convergence plots in Fig. 6e, the Significance Analysis of Microarrays (SAM) was conducted using the siggenes R package (v.1.68.0, delta threshold = 2), contrasting patients of the core glycolytic/plurimetabolic (GPM) and the core proliferative/progenitor (PPR) subtypes, according to Garofano et al. GBM pathway-based classification[23].

## Validation of cell-state modifiers from CRISPRa screen

For the functional validation of the candidate CRISPRa screen hits, IDH-wildtype-hGICs harboring a constitutively active dCas9-VPR system and the MGT4 mesenchymal sLCR were lentivirally transduced with constructs bearing four individual multiplexed sgRNAs targeting the respective screen hits, alongside a tag-BFP selection cassette for assessment of the transduction efficiency[45]. The levels of MGT4-reporter activation upon the CRISPRa-mediated transcriptional amplification of the candidate phenotypic drivers were assessed via FACS at 6 (early time-point) and 14 (late time-point) days post-transduction. At the experimental endpoint, the guide-bearing reporter-high-expressing cells were FACS-purified for subsequent whole transcriptome profiling.

## Transcriptomic profiling

The total RNA extraction from IDH-wildtype-hGICs-MGT4-dEGFP dCas9-VPR cells was performed using the TRIzol™ Reagent (Invitrogen, 15596026), followed by isopropanol precipitation and subsequent AMPure XP beads (Beckman Coulter, #A63881) purification. The RNA quantification and integrity assessment were conducted using the Qubit RNA High Sensitivity Assay Kit (Invitrogen, #Q32855) and the High Sensitivity RNA ScreenTape system (Agilent, #5067-5579), respectively. Multiplexed 3'-cDNA libraries were constructed starting from 60 ng of total RNA per sample as input, utilizing barcoded oligo-dT primers[69] in an adapted version of Bulk RNA barcoding and sequencing (BRB-seq) protocol[70]. The quantification of the final multiplexed library pools was performed using Qubit dsDNA HS Assay (Invitrogen, #Q32854) and Collibri™ Library Quantification (Invitrogen, #A38524500) kits. The library fragment size distributions were assessed using the TapeStation High-Sensitivity D1000 ScreenTape system (Agilent, #5067-5584). The sequencing of the multiplexed libraries was conducted on a NovaSeq 6000 in paired-end dual-index mode (Read 1: 28 bp, Index i7: 10 bp, Index i5: 10 bp, Read 2: 90 bp). Illumina index demultiplexing was performed using the bcl2fastq conversion software (v2.20.0). cutadapt (v2.1) was next used to extract the oligo-dT barcode sequences from Read 1, for downstream internal demultiplexing of reads using BRBseqTools-1.6.jar (http://github.com/DeplanckeLab/BRB-seqTools). The aligning of the demultiplexed data to a custom genome (MGT4-containing GRCh38) was performed using

STAR (v2.7.8a), and HTSeq (v2.0.2) was subsequently used to generate read count matrices.

The generated count matrices were analyzed using the DESeq2 R package (v.1.34.0). Briefly, the raw count data was converted into a DESeqDataSet with a design contrasting the CRISPRa-target-upregulated conditions against the parental unsorted control cells. The dataset was filtered for genes with more than 20 reads in at least three samples. Genes with absolute log2-fold change |>0.5| and adjusted p-values < 0.05 were considered to be significantly differentially expressed. Ensembl ID were mapped to gene symbols using EnsDb.Hsapiens.v79 (v2.99.0). log2FC-values derived from the differential comparisons between target gene overexpressing MGT#4-high and the control cells were used as pre-ranked input for conducting the fast gene set enrichment analysis (fGSEA) with the fgsea R package (v. 1.20.0) in Figure S7i.

## Signature scoring on single-cell RNAseq datasets

From a pan-GBM single-cell RNAseq study[24], we downloaded the core_GBmap, which consists of a harmonized single-cell RNAseq dataset stemming from 16 different studies and including a total of 338,564 cells from 110 patients. From the overall dataset, we extracted the malignant cell compartment based on metadata annotations. To establish module scores of each cell for a list of patient-derived GBM cell state signatures, sLCR target-gene signatures or single genes, we used the AddModuleScore() function and for visualization the FeaturePlot() function from Seurat (v.4.3). The pairwise correlation matrix between each module score was calculated and visualized using the corrplot package (v.0.92).

## Cell surface marker staining

IDH-wildtype hGICs stably transduced with MGT4 were harvested from suspension cultures. The cells were centrifuged at 300 × g for 5 min and resuspended in Accutase. After subsequent incubation at 37 °C for 5 min, a ten-fold excess of PBS was added and cells were centrifuged at 500 × g for 5 min. The harvested cells were aliquoted up to 1 ×10e6 cells/100 µL into FACS tubes and then subjected to staining with either primary biotinylated mouse anti-human CD44-antibody (BD Biosciences #555477; 1:50) together with secondary Streptavidin APC-eFluor 780 (Invitrogen #47-4317-82; 1:100) or primary conjugated mouse anti human CD24-Vio-Blue (Miltenyi Biotec #130-126-026; 1:50) and incubated for 30 minutes at room temperature in the dark. Unbound antibody was removed by washing the cells with 2 mL of Flow Cytometry Staining Buffer. This washing step was repeated twice. If an unconjugated primary antibody was used, incubation with an appropriate secondary antibody occurred after the primary antibody step. Finally, the cells were resuspended in 400 µL of Flow Cytometry Staining Buffer for flow cytometric analysis using a BD LSR Fortessa and appropriate laser/filter combinations for detection of cell surface marker staining and MGT4-d2EGFP expression.

## Statistics & reproducibility

For multiple comparisons of two groups, unpaired two-tailed Student's t test was used, unless otherwise specified. For comparisons of two or more groups, one-way ANOVA followed by Dunnett's post-hoc multiple comparisons test correction. For correlation analyses, Pearson correlation coefficients were calculated. Hierarchical clustering used Manhattan distance calculations. For boxplot representations, data distribution is shown with box indicating the interquartile range and inner line indicating the median. Whiskers extend to represent the data range, including outliers. Barplot data is shown as mean value +/− standard deviation. All experimental data has been derived from at least three independent biological replica, except for sequencing of the CRISPRa screen. No data were excluded from the analyses and the investigators were not blinded to allocation during experiments and outcome assessment.

## Reporting summary

Further information on research design is available in the Nature Portfolio Reporting Summary linked to this article.

## Data availability

All relevant data supporting the key findings of this study are available within the article and its Supplementary Information files. The sequencing data generated in this study have been deposited in the Gene Expression Omnibus database. Access can be obtained under the accession code "GSE236153". The processed sequencing data are provided in Supplementary Dataset S2. Source data are provided in the Source Data file. Source data are provided with this paper.

## Code availability

The LSD algorithm is available at: https://gitlab.com/gargiulo_lab/sLCR_selection_framework.

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

## Acknowledgements

We are grateful to L. Li, H. Naumann, M. Grossman and the MDC FACS and genomics technology platform for technical support and A. Akalin, M. Amendola, and S. Mzoughi for critical reading of the manuscript. The hCRISPRa-v2 library and SP-dCas9-VPR were a gift from J. Weissman (Addgene ID # 83978) and G. Church (Addgene plasmid # 63798). MDA-231, THP-1, NIH3T3 and CHO-K1 cell lines, were gifts from Rene Bernards (NKI, Amsterdam), S. Minucci (IEO, Milan), Italy), A. Hammes and T. Willnow (MDC, Berlin), respectively. Data analyses include data generated by the TCGA Research Network: https://www.cancer.gov/tcga. C.C., M.J.S. and Y.D. are graduate students with Humboldt University, B.J. and S.K. with Charitè Medical University. S.K. and Y.D. are affiliated with the Berlin School of Integrative Oncology (BSIO) at Charitè. The G.G. lab acknowledges funding from MDC, Helmholtz (VH-NG-1153), ERC (714922) and DFG (DFG SE 2847/2-1 to M.S.).

## Author contributions

C.C. developed the LSD algorithm and performed computational analyses in Figs. 1, 3, 4, S1, S3, S5, S6. Y.D. performed computational analyses in Figs. 4, 6, S4, S7, conducted and analyzed the CRISPRa screen and its validation and provided computational support for LSD pipeline curation and extension. M.J.S. performed computational and experimental analyses in Figs. 2, S2, S3f, 5, generated sLCR-bearing cell lines and performed FACS experiments. M.S. generated stable cell lines and reagents for Figs. 5, 6, S5, S7. Y.D., M.S. and M.J.S. conducted the CRISPRa screen and analyzed experimental data. B.J. conducted the sLCR-transfection screen in Fig. 2 and contributed to conceptual validation. S.K. contributed experimental support for conceptual validation. A.A. and J.Y. contributed unpublished reagents for the CRISPRa screen validation. I.B. developed the first generation of the sLCR pipeline and co-supervised the implementation of LSD. G.G. developed the concept, designed and supervised the study, secured funding together with M.S., interpreted the data and wrote the initial manuscript with inputs from C.C., Y.D., M.J.S., M.S. and I.B. for editing and revisions.

## Funding

## Competing interests

G.G. reports a patent application EP18192715 by the Max-Delbrück-Center for Molecular Medicine (MDC), Robert-Rössle-Str. 10, 13092 Berlin, Germany. No disclosures are to be reported by the other authors.
