## [Peer Review File · Nature Communications]

Logical design of synthetic cis-regulatory DNA for genetic tracing of cell identities and state changesReviewer #1 (Remarks to the Author):

This is a very interesting manuscript that follows on the outstanding work from the Gargiulo's laboratory to establish cis-regulatory DNA elements for genetic tracing of cellular states. In particular, the present manuscript reports the logical design of synthetic cis-regulatory DNA (LSD) framework as an experimentally validated computational tool that will streamline the tracing of cell identities and state changes for complex phenotypes by synthetic genetics. The value of LSD is especially high as it generates functional reporters without minimal promoters, a key advantage to increase phenotype specificity. In this manuscript, the authors show that it outperforms older generations of synthetic reporters for the tracing of the mesenchymal phenotype of glioblastoma. The new algorithm has been publicly released and will be very useful to the broad research community seeking to establish robust tracing of cancer cellular states. Whereas further improvements of the computational framework will unquestionably be possible, it will be exciting to see how the framework will allow the tracing of new pathway-inferred cellular states in cancer cells.

Reviewer #2 (Remarks to the Author):

The paper presents a framework for constructing synthetic regulatory regions to drive the expression of reporters or other cassette in cell-types or cell-states of interest. The method, logical design of synthetic cis-regulatory DNA (LSD), takes gene expression data and list of known TFs with PWM that characterize binding specificity. The method performs a search and outputs regulatory regions and region combinations hypothesized to drive expression in the target cell-state. The method can be extended to incorporate 3D genome organization and other data.

The computational procedure is straightforward and makes sense but is not in itself particularly exciting or novel. The exciting part about the paper is that the method appears to work in validation examples presented in the paper. The paper takes on a very important and timely question relevant for gene therapy, synthetic biology, and cell engineering. We have a very limited set of cell-type specific regulatory elements and a major extension in number across cell types would significantly boost current AAV research and immune cell engineering. I think the main issue is validation; validation feels quite limited and focused on cases the authors appear to know well. The authors primarily validate their strategy through limited experiments here in specific glioblastoma cell lines and also apply a CRISPRa screen that seems somewhat circuitous to me as a validation.

From the perspective of this reviewer, the paper would be a major advance if the authors were able to validate expression in a series of mouse or human cell-types—even focusing on cell-types that can be transduced easily. It would be very interesting if the authors could generate, for example, synthetic sequences that can drive expression in a basket of standard cell-lines: Jurkat cells, immortalized neural stem cells, embryonic stem cells, epithelial cell lines— or even a basket of immortalized cancer cell lines from breast, prostate, etc . Could they even design sequences that can fit within AAV packaging limits and drive expression in specific primary cell-types of interest?

A home run would be more comprehensive validation in mouse using Tabula Munis data as an input.

Currently, I don't feel that the paper warrants publication in Nature Communications. Validation on a broader basket of cell types would significantly strengthen the findings of the paper and also enable a better assessment of how general and natural the method is to apply outside of situations the authors have studied in detail below.

Clearly expression and transduction and access or important constraints. I think a reasonable validation would involve 4-5 cell-types that span a set of different underlying tissue types.

Reviewer #3 (Remarks to the Author):

The authors developed an automated pipeline for creating cis regulatory elements based on the enrichment of transcription factor binding sites. However, neither the novelty nor the validation suffices for publication in this journal. There's little algorithmic innovation. The pipeline simply automates something that human users can already manual curate using a greedy search. We know that TF regulation is highly context dependent. It is not clear whether using CREs from the author's greedy search would even outperform using multiple synthetic copies for each TFBS, and there are no analysis or data in that regard. On the validation side, much more experimental data are required to benchmark the new regulator elements against canonical markers for cell types/states, and very few future users will be willing to adopt this pipeline if it is not convincingly demonstrated that the de novo elements are not only functional but also specific. Without characterizing the enhancers in cell types/states that might share many TFs, the self-referring validation against the pre-defined TF sets does not shed much light on the actual performance of the designed elements, and there's little evidence to support the use of these elements in a manner implied in the title.

REVIEWER COMMENTS

Reviewer #1 (Remarks to the Author):

This is a very interesting manuscript that follows on the outstanding work from the Gargiulo's laboratory to establish cis-regulatory DNA elements for genetic tracing of cellular states. In particular, the present manuscript reports the logical design of synthetic cis-regulatory DNA (LSD) framework as an experimentally validated computational tool that will streamline the tracing of cell identities and state changes for complex phenotypes by synthetic genetics. The value of LSD is especially high as it generates functional reporters without minimal promoters, a key advantage to increase phenotype specificity. In this manuscript, the authors show that it outperforms older generations of synthetic reporters for the tracing of the mesenchymal phenotype of glioblastoma. The new algorithm has been publicly released and will be very useful to the broad research community seeking to establish robust tracing of cancer cellular states. Whereas further improvements of the computational framework will unquestionably be possible, it will be exciting to see how the framework will allow the tracing of new pathway-inferred cellular states in cancer cells.

*We are very much grateful to the Reviewer for their enthusiastic take on our manuscript and for inspiring the incorporation of a straightforward analysis that we believe it increased the value of our example phenotypic target discovery by combining LSD with patients' expression profiles differential gene expression and pathway analyses as well as patients' derived primary cells' vulnerability maps. In our original manuscript, we had focused on the Verhaak classification and overlooked the exciting pathway-based glioblastoma classification from Garofano et al. Nature Cancer 2021. We have now reanalyzed the TCGA GBM microarray data (**Figure 1 – R1**) and incorporated the pathway-based classification by Garofano & Iavarone (Nature Cancer 2021) to demonstrate how LSD enables the tracing of new pathway-inferred cellular states in cancer cells. We agree that the pathway-based classification is a highly informative alternative to more broadly used bulk-/sc-RNA-seq classifications and have combined it with other datasets, including the Richards & Dirks dependency map (Nature Cancer 2021) and our CRISPR-activation screen data with 3 independent reporters. Our previous approach identified experimentally validated FOSL1 and WWTR1 among a shortlist of mesenchymal GBM drivers, and with the integration of the Garofano data, we find that that the mesenchymal GBM and the GPM GBM classification are distinct, albeit convergent in some aspects. Notably, FOSL1 and WWTR1 appear to drive the convergent portion of the Mesenchymal-Glycolytic/Plurimetabolic identity (**new Fig. 6f**). We believe will strengthen the interest in deeply characterizing the identity of glioblastoma cells and makes it straightforward and exciting to apply LSD to the specific tracing of new pathway-inferred cellular states in cancer cells, as suggested by the R1.*

Figure 1 - R1 | Gene expression-based classification of TCGA microarray data.

A) Distribution of log₂ lowess normalized sample/reference signal (cy5/cy3) ratios of TCGA-GBM patients profiled with Agilent 244K Custom Gene Expression array (G4502A_07_1/2).

B) Principal component analysis of the core TCGA-GBM patients subtyped according to the GBM pathway-based classification (Garofano et al.).

C) Heatmap visualization of the Gene Set Variation Analysis (GSVA) enrichment scores for the indicated gene sets in the TCGA-GBM expression data.

We have also spontaneously added a minimal set of experimental validation data, such as driving MGT4 activation through endogenous activation of RELA, RAC1 and FOSL1, using 4 independent guide RNAs and performed target confirmation and GSEA using RNA-seq (Fig. s7g-i). Together, the new data resulted in a few additional panels in the current Figure 6 and Figure S7 (formerly Fig. 5/s5).

We hope that our spontaneous analyses and validation are aligned with the expectations from this Reviewer, but we also appreciate the R1 for having opened to our manuscript be published as is in Nature Communications, which is a rare event.

Reviewer #2 (Remarks to the Author):

The paper present a framework for constructing synthetic regulatory regions to drive the expression of reporters or other cassette in cell-types or cell-states of interest. The method, logical design of synthetic cis-regulatory DNA(LSD), takes gene expression data and list of known TFs with PWM that characterize binding specificity. The method performs a search and outputs regulatory regions and region combinations hypothesized to drive expression in the target cell-state. The method can be extended to incorporate 3D genome organization and other data.

The computational procedure is straightforward and makes sense but is not in itself particularly exciting or novel. The exciting part about the paper is that the method appears to work in validation examples presented in the paper. The paper takes on a very important and timely question relevant for gene therapy, synthetic biology, and cell engineering. We have a very limited set of cell-type specific regulatory elements and a major extension in number across cell types would significantly boost current AAV research and immune cell engineering. I think the main issue is validation; validation feels quite limited and focused on cases the authors appear to know well. The authors primarily validate their strategy through limited experiments here in specific glioblastoma cell lines and also apply a CRISPRa screen that seems somewhat circuitous to me as a validation.

We would like to thank this Reviewer for highlighting that the area in which we operated is “very important and timely” and for defining “exciting that our the method appears to work in validation examples presented in the paper”. We agree with your constructive comments and with assessment that validation is a crucial aspect of our work. We have taken your suggestions as a blueprint to strengthen the findings and broaden the scope of our study.

We also much appreciate the R2’s reasonable approach to challenge us to venture outside of the proneural-to-mesenchymal transition in glioblastoma so that our readers would be convinced that the method is robust. Not only we agree that this is an important evidence for us to provide, but in fact, we had not submitted this manuscript until we had that evidence for the SARS-CoV-2 application of LSD. This manuscript is now published in Science Advances (Jiang B., Schmitt M.J., et al, 2023).

Whereas we had attached the preprinted manuscript to the original submission to make sure that our method appeared robust beyond the cases offered in the manuscript, we appreciate that the suggested line of validation by this Reviewer would appeal specifically to scientists in the gene therapy space, and therefore we generated short and robust AAV-compatible sLCRs for gene therapy applications (see below).

From the perspective of this reviewer, the paper would be a major advance of the authors were able to validate expression in a series of mouse or human cell-types—even focusing on cell-types that can be transduced easily. It would be very interesting if the authors could generate, for example, synthetic sequences that can drive expression in a basket of standard cell-lines: Jurkat cells, immortalized neural stem cells, embryonic stem cells, epithelial cell lines— or even a basket of immortalized cancer cell lines from breast, prostate, etc . Could they even design sequences that can fit within AAV packaging limits and drive expression in specific primary cell-types of interest?

A home run would be more comprehensive validation in mouse using Tabula Munis data as an input.

Currently, I don't feel that the paper warrants publication in Nature Communications. Validation on a broader basket of cell types would significantly strengthen the findings of the paper and also enable a better assessment of how general and natural the method is to apply outside of situations the authors have studied in detail below.

Clearly expression and transduction and access or important constraints. I think a reasonable validation would involve 4-5 cell-types that span a set of different underlying tissue types.

To address your concern regarding validation in a broader range of cell types, we have designed eight "housekeeping sLCRs" that are compatible with AAV promoters for gene therapy applications. We utilized Tabula Sapiens and Tabula Muris data, as well as bulk and single-cell ATAC-seq data, as input to generate these sLCRs.

We compared their performance to the shortest and most potent promoter compatible with AAV genome size, namely the EFS short version of the EF1A promoter. We much appreciated the reasonable consideration by this Reviewer that attempting in vivo validation would have been beyond the scope and the timeline of this revision and we followed the advice to validate these reporters in cell lines. To that end, we generated 54 individual stable cell lines, encompassing human blood, brain, breast and kidney tissues of origin, as well as rodent species (mouse and hamster). Our housekeeping sLCRs consistently produced signal above background, and in some cases, their performance was comparable to or even better than the EFS promoter in terms of strength and low variation across cell lines. We have included the results of these experiments in two new figures (Fig. 5 and Fig. s5) and provided supplementary table s1 to support our findings.

Furthermore, we took on the challenge of designing kidney-specific sLCRs compatible with AAV genome size. Given the sparse nature of Tabula Sapiens and Tabula Muris data, we curated multiple available datasets to obtain robust biomarkers and also incorporated kidney-specific scATAC-seq data as a reference for enriching tissue-specific cis-regulatory elements (CREs). As also highlighted by the Reviewer, an experimental validation of this complex endeavor is beyond the scope and timeline of this revision, because of the transduction and in vivo components of such studies. Nevertheless, we conducted in silico testing of the designed kidney-specific sLCRs against sLCRs designed for human lung and liver tissues, which supported specificity. If the in silico validation is considered sufficient by Editors and Reviewers, we will gladly report these findings in Fig. s6 and supplementary table s1, providing a proof-of-principle for designing tissue-specific and AAV-compatible sLCRs.

In the future, we have also established an internal MDC collaboration with an expert colleague (Michael Kaminski) with whom we plan to increase the specificity and granularity of our sLCRs, in order to reach mouse and human specific kidney cell types.

Overall, we much welcome the feedback from this Reviewer, whose comments helped us to extend the applicability of LSD to gene therapy with a specific focus. We hope to have now gained full support for publication of LSD in Nature Communications.

Reviewer #3 (Remarks to the Author):

The authors developed an automated pipeline for creating cis regulatory elements based on the enrichment of transcription factor binding sites.

However, neither the novelty nor the validation suffices for publication in this journal. There's little algorithmic innovation. The pipeline simply automates something that human users can already manually curate using a greedy search.

We appreciate the opportunity to respond to the comments made by this Reviewer.

On the innovation of the algorithm, we believe to be well positioned to judge the extent of automation vs innovation of the algorithm because we have developed the original one. In Schmitt, Company, Dramaretska et al., Cancer Discovery 2021, the first generation of the algorithm limited its output to the matrix of 150bp candidate CREs and the TFBS potentially binding these. As a result, users needed to make decisions on which elements to prioritize and how to build the definitive sLCR. This could (1) be user-dependent and (2) result in different outputs depending on users' knowledge of transcription and regulatory networks for the target phenotype. LSD automates the first steps and – on the top of that – it provides a single bona fide output for each input. By comparing side-by-side LSD-designed MGT4 and the “user-dependent” MGT1 that was validated in silico, in vitro and in vivo, we show that such an important computer-assisted decision (i.e. which elements to prioritize, and how to assemble the final sLCRs) indeed result in a functional and specific sLCR. Furthermore, we incorporated the possibility to focus on users-defined reference genomes, such as chromatin accessibility and 3D contact maps, that enables increasing the specificity of the output. As inventors of the approach, for us, all of this was a game changer because we could venture in more applications with limited prior knowledge but with some degree of confidence that a carefully curated input would determine a solid output. We hope that our explanation gives to the Reviewer a different perspective on how adding a few steps to the original code, along with validating its performance and establishing ground rules, together constitute an important advance, as appreciated by both our R1 and R2.

In case we have missed a reference to any other software doing similar tasks, we would much welcome the input from Reviewers. We would gladly acknowledge prior work that contributes to innovation in this space, we simply couldn't find it.

R3: We know that TF regulation is highly context dependent. It is not clear whether using CREs from the author's greedy search would even outperform using multiple synthetic copies for each TFBS, and there are no analysis or data in that regard.

We have not claimed that our approach outperforms the use of 'multiple synthetic copies for each TFBS. To our knowledge, building an entire mammalian promoter from an ensemble of TFBS, functional and specific without minimal promoters from non-specific settings, has not been systematically done. Nevertheless, we believe that the two approaches serve different purposes.

LSD enables users to transform a transcriptional phenotype, or a mixed proteomic-transcriptional phenotype, into a synthetic cis-regulatory cassette most likely active in that context and potentially specific to it.

Limited to our knowledge, using “multiple synthetic copies for each TFBS” has been primarily used to study well defined cis-regulatory circuits in isolation. We are familiar with the pioneer

work from the Elowitz and Wendell labs, to name some high profile labs operating in the space of studying synthetic circuits, and we do not see LSD as an alternative to using specific TFBS to focus on specific pathways. However, comprehensive knowledge of all pathways active in various cell identities, and how interconnected they are, may be beyond the assets of the average user. It certainly is for us, in most cases we have studied. In this regard, we believe LSD offers a unique opportunity to build synthetic CREs building on reasonable assumptions which aiming at cell identity specificity.

Inspired by the comment of this Reviewer, we added one passage to our discussion suggesting that “to aim at precise cell-type targeting within a tissue, it may be advantageous to combine LSD as a blueprint for designing tissue-specific promoters and reinforce cell-type specificity by incorporating cell-type-specific TFBS, which has proven to be an effective strategy for focusing on specific transcriptional programs ⁴⁷⁻⁵⁰.”

Hence, we are grateful for the input.

R3: On the validation side, much more experimental data are required to benchmark the new regulator elements against canonical markers for cell types/states, and very few future users will be willing to adopt this pipeline if it is not convincingly demonstrated that the de novo elements are not only functional but also specific.

Without characterizing the enhancers in cell types/states that might share many TFs, the self-referring validation against the pre-defined TF sets does not shed much light on the actual performance of the designed elements, and there's little evidence to support the use of these elements in a manner implied in the title.

GG: Inspired by the comment of this Reviewer, we selected CD24 and CD44 as “canonical markers for cell types/states”. CD24 and CD44 were used in the influential paper by Neftel et al. Cell 2019, as proxy of non-mesenchymal and mesenchymal glioblastoma cell states (Franca et al., bioRxiv 2022). They were also used as proxy for more epithelial and drug sensitive vs more mesenchymal and drug resistant ovarian cancer cell states, thereby featuring a good benchmark to test how our sLCRs would perform against these markers. First, we exploited an ensemble of single-cell studies in glioblastoma, that represented a sufficiently large dataset to test hypotheses. To that end, we used the ~1M cells integration by Ruiz Moreno et al. (bioRxiv 2022). We show that CD24 and CD44 overlap to some extent, overlapping with Neftel NPC and mesenchymal states, respectively (new Fig. s2f). Using unbiased correlation of meta modules scores, we also show that both CD44 and MGT4 cluster with mesenchymal gene sets, whereas both CD24 and PNGT3 cluster with proneural/NPC gene sets, albeit PNGT3 appears to better cluster with those than CD24 (new Fig. 2c). Experimental results show that MGT4 exhibits higher sensitivity than CD44, as it is upregulated in a cellular model for PMT, whereas CD44 is not (new Fig. 2d). We previously showed TNF α to drive a gene expression program overlapping with that of mesenchymal patients from TCGA cohort in our model (Schmitt, Company, Dramaretska et al., Cancer Discovery 2021), and our data clearly show that MGT4 is induced significantly in response to TNF α , whereas CD44 is not. CD24 marginally but significantly increases during a process that is clearly PMT, instead of doing the opposite or being stable, highlighting that relying solely on each marker as a proxy for proneural and mesenchymal GBM identity is insufficiently predictive.

Nevertheless, in our view, the most important piece of evidence in our new **Fig. 2c**, is that CLGT3 (and ACGT1/3) sLCRs cluster with classical/AC-like glioblastoma meta-modules. Regarding these identities, we are not aware of consensus around a canonical marker, thereby featuring the most important asset of LSD: the ability to design markers for novel or very specific cell identities. We believe this analysis further substantiated the utility of sLCRs for tracing complex phenotypes, as described in our previously published work (Schmitt, Company, Dramaretska et al., *Cancer Discovery* 2021; Serresi et al., *Sci.Adv.* 2021; Jiang, Schmitt et al. 2023, *Sci.Adv.*).

We hope that our rebuttal, the new data/discussion inspired by the Reviewer, together with those inspired by the other reviewers and the independent paper published in *Science Advances* by applying LSD to SARS-CoV-2 infected epithelial cells, will constitute ground for this Reviewer to now support publication of LSD in *Nature Communications*.

Reviewer #1 (Remarks to the Author):

The authors have addressed all my concerns.

Reviewer #2 (Remarks to the Author):

Dear Authors,

I have been through the revised manuscript many times now. While I applaud the authors for introducing Figure 5, I have had an enormous amount of difficulty understanding exactly what experiment was done and what is being shown in this figure. I think Figure 5 is very important for validating the broader claims made in the paper.

Focusing on Figure 5, the authors say in the text that they generated "54 individual stable cell lines, encompassing human blood, brain, breast and kidney tissues of origin, as well as rodent species.

Again-- this is a laudable experiment. What I am having a very difficult time interpreting-- is the data.

To evaluate the performance of LSD, I would have liked to see two types of plots:

(1) Histograms/violin plots of reporter gene expression across all cell lines and control cell lines-- with very clear labels. I don't think this is in S5. In general the author captions-- both S5 and Figure 5 are very short and difficult to interpret.

It is important for us to see the population structure of the cell populations with LSD designed promoters. Is expression uniform? How does it vary between cell lines? Are all promoters equally good-- are some tissues of origin more difficult?

It is difficult for me to evaluate the claims without having a more detailed and more clear version of Figure 5. I think this is important because generality is a major claim of the paper.

(2) Analysis of expression relative to endogenous genes expressed in some of the cell lines. This could be done for a sub-set of the cell lines. A really beautiful experiment would be to compare single-cell mRNA-seq for endogenous gene expression and LSD designed promoter reporters. But I could also imagine doing something easier like immunology fluorescence.

I think Figure 5 is central to the case for this paper being published. I would recommend caution without a more detailed presentation and explanation of the data. The current Figure 5 seems very difficult to interpret.

Reviewer #3 (Remarks to the Author):

I appreciate the author's additional efforts. They are nontrivial and have certainly strengthened the paper. I'm encouraged by their new results regarding CD24 and CD44, which definitely points in the right direction. However, I still feel uneasy about two aspects: comparison to 1st gen and specificity.

1. The authors doubled down on what an improvement their automation is compared to the more manual 1st gen. However, their rebuttal also says "By comparing side-by-side LSD-designed MGT4 and the "user-dependent" MGT1... we show that such an important computer-assisted decision... indeed result in a functional and specific sLCR." It's curious that they didn't say "...result in a better sLCR than MGT1", and Fig. 2b suggests that the reason might be MGT4 is NOT better than MGT1. What would it look like if MGT1 were included in Fig. 2c and 2d? It seems too obvious an omission and makes one wonder. Furthermore, if MGT4 is the focus, wouldn't it be more straightforward to show lenti vs piggyBac for MGT4 in Fig. 2b, rather than inferring that from MGT1?

2. The bigger issue is specificity, which is referred to many times throughout the paper. It is therefore very curious that, when prompted by another reviewer, the authors decided to spend their effort on experimentally testing housekeeping reporters and the "specificity" results are only in silico. As the other reviewer pointed out, the major appeal of the paper would be "We have a very limited set of cell-type specific regulatory elements...". Unfortunately, with the exception of Fig. 2c, 2d, the reviewer's comment that "validation feels quite limited and focused on cases the authors appear to know well." remains the case. Unless the other reviewer makes a much more positive assessment, I have the feeling that the author's did a bait-and-switch, implying specific reporters and delivering mostly housekeeping ones, and my concern has remained regarding how well their method performs beyond glioblastoma. The authors seem totally capable of making and experimentally testing the same number of reporters in the same number of cell lines, not housekeeping but cell-type-specific ones. Why they chose not to do so is beyond me.

Point-by-point Response

Reviewers' comments:

Reviewer #1 (Remarks to the Author):

The authors have addressed all my concerns.

Once again, we greatly appreciate the valuable feedback by this Reviewer.

Reviewer #2 (Remarks to the Author):

Dear Authors,

I have been through the revised manuscript many times now. While I applaud the authors for introducing Figure 5, I have had an enormous amount of difficulty understanding exactly what experiment was done and what is being shown in this figure. I think Figure 5 is very important for validating the broader claims made in the paper.

Focusing on Figure 5, the authors say in the text that they generated "54 individual stable cell lines, encompassing human blood, brain, breast and kidney tissues of origin, as well as rodent species.

Again-- this is a laudable experiment. What I am having a very difficult time interpreting-- is the data.

We value your thorough review and your emphasis on the significance of Figure 5/s5 for extending our paper's claims to a broader readership.

We followed your instruction towards expanding the scope of LSD (i.e. we tackled one challenge in AAV-gene therapy), of generating and validating new reporters (i.e. we made 8) in 4-5 cell lines (we made 6), without biases in species or tissues. We were also tasked with using "Tabula Muris" as input and we did use both *Tabula Muris*, *Tabula Sapiens*, as well as tissue-specific scATAC-seq data.

If we have been parsimonious in the text and legends, we regret that. We have therefore entirely rewritten that paragraph and appreciate the opportunity to offer additional clearer explanations here. In turn, this will facilitate the interpretation of the extensive amount of novel computational and experimental data that we produced to address the main concerns raised.

To evaluate the performance of LSD, I would have liked to see two types of plots:

(1) Histograms/violin plots of reporter gene expression across all cell lines and control cell lines-- with very clear labels. I dont think this is in S5. In general the author captions-- both S5 and Figure 5 are very short and difficult to interpret.

It is important for us to see the population structure of the cell populations with LSD designed promoters. Is expression uniform? How does it vary between cell lines? Are all promoters equally good-- are some tissues of origin more difficult?

it is difficult for me to evaluate the claims without having a more detailed and more clear version of Figure 5. I think this is important because generality is a major claim of the paper.

Again, if the legends were not clear, we apologize. We have now increased the detail in the schematics in Figure 5a and used more words to enhance clarity. But we would like to stress

that the data the Reviewer is herewith asking for was already in Fig.5c, Fig.5d and Fig. s5c, yet apparently hard to grasp.

Below, please find the individual panels and the breakdown of the specific requests.

The box plot in Figure 5c depicted the expression of mCherry (e.g., the fluorescent reporter) in each of the cell lines (i.e., individual dots) that were transduced in triplicate with each of the housekeeper-like sLCRs generated (i.e., third box), as compared to the most powerful short and AAV-compatible housekeeping promoter known to us (i.e., EFS, second box).

The 1st box (labeled “Control”) shows the background expression of mCherry in unmodified parental cell lines from various tissues and species (in triplicate). Based on statistical comparison, the simplest interpretation of this data are that:

1. we obtained 100% success in generating functional promoters (please compare column 3 “HKsLCR” to column 1 “Control”).
2. The generated housekeeper-like reporters display a variable level of expression as compared to the most powerful short and AAV-compatible housekeeping promoter know to us (i.e. EFS, please compare 2nd and 3rd boxes).

We decided against the violin plot because the box plot is a better representation when the scope is comparing median and the numbers are limited to tens rather than hundreds.

Next, we looked into the observation (2.) from Figure 5c. To do this, we created a heatmap plot in Figure 5d that shows how well each housekeeper-like sLCR performed in different cell lines. We did triplicate measurements of each sLCR in each cell line and normalized it to the parental non-transduced cell line. The resulting mCherry intensity values are visualized via the color-coding scale on the left, while the sample and sLCR position is determined by hierarchical clustering (as seen in the dendrograms in Figure 5d's heatmap to the left and upper sections). This helped us tell apart the housekeeper-like sLCRs that worked similarly to the EFS promoter from the ones that weren't as strong. The colors and labels in the heatmap make it clear which promoters grouped with EFS (like HKGT1, HKGT4, and HKGT4-short) and why (for instance, the average expression in each cell line, shown in the column).

In order to provide a detailed view of the performance of each vector in each cell line, in Figure S5c, we had also provided a distribution and individual data points (lines) per cell line (color).

In Figure 5e (above), the statistical analysis examines data dispersion categorized by promoter type. The bubble size denotes standard deviation (i.e. the lower, the better), the x-axis denotes intensity values (i.e. the higher, the better), and the y-axis denotes the interquartile range (IQR) on variability, representing the range between the 25th and 75th percentiles (i.e. the lower, the better). We have explored diverse metrics of data variability, and this comprehensive data is elaborated in Figure S5d and Supplementary Table S1 – Tab 49. Our analyses uncover that while the EFS promoter showcases great potency it also demonstrates heightened variability across all cell lines. This observation underscores that HKGT4 and HKGT4s – within the context of the cell lines tested in this experiment – outperform EFS in terms of variability, a pivotal criterion in defining a "housekeeper" promoter.

Overall, the data generated within the framework of a revision plan tailored to the instructions by this reviewer, within the reasonable timeframe of such a revision, support our conclusion that we succeeded in generating housekeeper-like sLCRs compatible with size constraints of AAV vectors (e.g. 100%), and particularly some showing potency that is quantitatively comparable to EFS, which was unparalleled for more than a decade (i.e. HKGT1, HKGT4 and HKGT4-short).

Although we tend to favour the current representation, to make sure that our data can be interpreted without hesitation, we herewith share with our Reviewers three additional representations of the distribution of the data in terms of:

- intensity/cell line (note the distribution of the cell lines within each violin)

- intensity/cell line/promoter (note all the reporters vs controls).

- overall intensity (note the distribution of the signal and outliers).

Moreover, for every reporter that we generated, we indicated the exact composition of the signature genes and TFBS (Table S1) and each figure will have "source data" for the readers to go through individual plots and conclusions associated with this.

We have now made sure that our text is unequivocally understood by a broad readership and we hope that our extended explanation and revised manuscript can secure the support of this reviewer.

(2) Analysis of expression relative to endogenous genes expressed in some of the cell lines. This could be done for a sub-set of the cell lines. A really beautiful experiment would be to compare single-cell mRNA-seq for endogenous gene expression and LSD designed promoter reporters. But I could also imagine doing something easier like immunology fluorescence.

We have compared side-by-side cell lines engineered with the exact same vector and wherein EFS was the chosen control. We believe that this is the most critical control.

Albeit we believe that connecting sLCR expression to endogenous markers is an additional request by the Reviewer and is incremental, we were willing to address the point experimentally.

The most accurate way to connect our transcriptional reporters' expression to endogenous genes is RT-qPCR. Thus, we measured the expression of established endogenous housekeepers that are part of the LSD input for this design (i.e. GAPDH and ACTB) and compared their expression to that of our reporters. To that end, we used primer sets which were pre-tested for linearity in RT-qPCR. We further corrected the expression of our synthetic housekeeping sLCRs for copy number integrations.

We have generated the data for the representative human cell lines tested in Fig.5 and for the house-keeping sLCRs with the highest activities across conditions (i.e. HKGT1, HKGT4, HKGT4s), alongside with the EFS promoter.

We have collected mRNA and gDNAs from all the cell lines generated and used defined nucleic acid equivalents as target for qPCR (10ng gDNA/well, 2ng mRNA/well, n=4). The **Figure 1 for Reviewers** shows that EFS, HKGT4 and HKGT4s display a comparable performance as endogenous housekeepers in all three cell lines, in particular to GAPDH. This also confirms

the FACS data previously showing that HKGT1 has lower expression levels compared to HKGT4/4s. We show that HKGT4 and HKGT4s are only slightly less expressed than the strong EFS promoter, after their expression level is normalized to correct for copy number integrations (e.g. by dividing mCherry mRNA copies by mCherry gDNA copies). This quantitative data also confirms that HKGT4s is less expressed in 8MGBA and MDA-MB-231 compared to HKGT4, due to the deletion of a 150bp CRE. This phenomenon is far less obvious in 293T, suggesting that TFBS in this particular CRE are specifically expressed in these cell lines but not 293T.

Figure 1 [for editors and reviewers only]: Comparing endogenous and synthetic housekeeper-like expression at quantitative mRNA levels. Bar plot illustrating the average relative mRNA quantities by one-step RT-qPCR for the indicated primer sets in the indicated cell lines. Delta-Ct values obtained for 2ng/well of total mRNA were normalized to a heterochromatic region (N2; surrogate for diploid copy number) in the case of GAPDH and b-Actin or HKSLCRs mCherry lentiviral integration sites amplified from genomic DNA (gDNA). Each bar represents a specific HKSLCR reporter, and the height of the bar

corresponds to the mean normalised mRNA expression. Additionally, one-sided error bars indicate the standard error of the mean. Data is organized in a grid by primers and cell lines with distinct colors.

Overall, as requested, the above data connected the expression of our housekeeper-like sLCRs not only to the synthetic EFS promoter, but also to endogenous genes. GAPDH appears a more stable housekeeper across the cell lines chosen, despite the limitation of copy number variations across cells and cell lines. We note that the Housekeeping and Reference Transcript Atlas concludes that housekeepers should be selected based on the cell/tissue model (e.g. there is no universal housekeeper gene, Hounkpe et al., 2021 *Nucleic Acids Research*), and these data do not challenge previous conclusions, but also do not add particular aspects to the data interpretation. Hence, in our view these should be retained at the rebuttal level.

I think Figure 5 is central to the case for this paper being published. I would recommend caution without a more detailed presentation and explanation of the data. The current Figure 5 seems very difficult to interpret.

We appreciate the statement by this Reviewer. From our point of view, Figure 2b is the central case for the paper being published as it shows that - in a complex task like the designing of a reporter for EMT in cancer - the computer can behave as the informed human, implying that it will be able to run similar tasks in other contexts as well. Such implication was validated in the accompanying paper by Jiang, Schmitt et al. 2023 *Sci. Adv.* where LSD could design a reporter specific for epithelial cells' responses to viral infections by an RNA virus.

Nevertheless, we agree that Figure 5 constitutes an additional direct validation and - as we successfully addressed an entirely new task requested by the Reviewer - the testing of LSD is more robust, and its use may be extended to one more case (along with additional reagents available to the community).

We hope that the increased level of detail provided in this rebuttal, in the revised manuscript and figure, help address the remaining concerns that data which address their questions (*Is expression uniform? How does it vary between cell lines? Are all promoters equally good-- are some tissues of origin more difficult?*) were already present and are now more clearly described.

Reviewer #3 (Remarks to the Author):

I appreciate the author's additional efforts. They are nontrivial and have certainly strengthened the paper. I'm encouraged by their new results regarding CD24 and CD44, which definitely points in the right direction.

We are glad that the Reviewer appreciates the data we provided in response to their specific concern.

We note that we also appreciated the specificity of the request (i.e. to "benchmark the new regulator elements against canonical markers for cell types/states").

However, I still feel uneasy about two aspects: comparison to 1st gen and specificity.

In this case, it was difficult for us to address an aspect mentioned now directly but not explicitly asked before, at least not by this Reviewer. We welcome the opportunity to address the specific critiques in their merit.

1. The authors doubled down on what an improvement their automation is compared to the more manual 1st gen. However, their rebuttal also says "By comparing side-by-side LSD-designed MGT4 and the "user-dependent" MGT1... we show that such an important computer-

assisted decision... indeed result in a functional and specific sLCR." It's curious that they didn't say "...result in a better sLCR than MGT1"

We respectfully note that the Reviewer must have missed that we have stated in the original & revised abstract that "A mesenchymal glioblastoma reporter designed by LSD outperformed previously validated ones [...]."

We have also specifically elaborated on this [page 5, lines 220-223]: "Interestingly, despite the fact that the MGT4 reporter was designed by LSD on a different TFBS list, it outperformed the first generation MGT1-2 on their specific TFBS input list (Fig. 3a)."

and Fig. 2b suggests that the reason might be MGT4 is NOT better than MGT1.

We argue that "better" must refer to a specific aspect. In terms of functionality and specificity, we have proven that MGT1 is indeed functional and specific in Schmitt, Company, Dramaretska, et al. (Cancer Discovery 2021, glioblastoma) and Serresi et al. (Science Adv. 2021, lung cancer). Hence, the scope of the experiment in Fig. 2b was to prove to ourselves and others that LSD can design a mesenchymal identity reporter as good as MGT1, or better. In the specific experiment presented in Fig.2b, wherein MGT1 and MGT4 measure a cell identity shift in response to TNF-alpha, given that the cells are the same and the signaling cue is the same, it may simply not be possible for MGT4 to outperform MGT1 and it is certainly not the scope of the test.

We show that MGT4 outperforms MGT1 in Fig.3a, 3c, s3a.

What would it look like if MGT1 were included in Fig. 2c and 2d? It seems too obvious an omission and makes one wonder.

It is rather surprising that the misunderstanding of the plot in Fig. 2c and the lack of duplication of Fig. 2b in Fig. 2d lead to questioning the entire ground of our work.

In Fig. 2c, which we made to address the request by this Reviewer to compare LSD to canonical markers, we focused on MGT4 because this is LSD-designed. However, as we clearly indicated in the text "The PNGT3, CLGT3 and MGT4 sLCRs were designed in an unbiased manner by LSD and their specific signature genes were identical to those of the first-generation sLCRs (Fig. S1a-b), while we defined the TF lists by differential enrichment (see Methods). [page 3, line 116-118]". Hence, MGT4 could also read MGT1/2/4, because MGT1, MGT2 and MGT4 have the exact same signature genes, which are the input for the correlation analysis (compare Fig. 2 Legend "c) Correlation plot between patient-derived glioblastoma cellular state signatures and module scores of sLCR signature genes for pan-GBM data from Ruiz-Moreno et al. (2022).").

We have now edited the legend to clarify this, which will certainly avoid jumping to conclusions, albeit we disagree that this is an obvious omission.

As per Fig. 2d, MGT1 was not previously incorporated in figure because we were asked by this Reviewer to compare a LSD-based reporter (i.e. MGT4) to "canonical markers". Please find here also MGT1 but note that we did not run the experiments at the same time for the above reasons, and we therefore we wish to show this in the rebuttal only, without adding it to the primary figure panel.

Furthermore, if MGT4 is the focus, wouldn't it be more straightforward to show lenti vs piggyBac for MGT4 in Fig. 2b, rather than inferring that from MGT1?

We believe the Reviewer has misunderstood the core set of comparisons that are shown in Figure 2b. As explained in the text and above, in Fig. 2b we needed to prove that MGT4 was functional and specific like MGT1, which was validated in our past research, and to do so in the same cellular model. MGT1 and MGT4 are experimentally compared head-to-head in the exact same piggyback vector, and therefore there is no inference from MGT1 in this comparison.

The lenti vs piggyback vector comparison is between MGT1 lenti (previously published in Schmitt et al., 2021 and Serresi et al. 2021), and available to the community through Addgene, and MGT1 piggyback, made for this paper. This supports our conclusion that "*LCRs' activity is mainly directed by the synthetic cis-regulatory DNA and largely independent of the genome integration bias of the vector system employed. [page 4, lines 140-143]*". Once again, this is an experimental head-to-head comparison, there is no inference here either.

2. The bigger issue is specificity, which is referred to many times throughout the paper. It is therefore very curious that, when prompted by another reviewer, the authors decided to spend their effort on experimentally testing housekeeping reporters and the "specificity" results are only in silico. As the other reviewer pointed out, the major appeal of the paper would be "We have a very limited set of cell-type specific regulatory elements...". Unfortunately, with the exception of Fig. 2c, 2d, the reviewer's comment that "validation feels quite limited and focused on cases the authors appear to know well." remains the case. Unless the other reviewer makes a much more positive assessment, I have the feeling that the author's did a bait-and-switch, implying specific reporters and delivering mostly housekeeping ones, and my concern has remained regarding how well their method performs beyond glioblastoma.

As explained clearly above, we believe to have addressed the specificity once the computer could design an MGT4 that performed as good or better than MGT1. We interpreted R2's comments as motivation to go beyond the field of GBM and proneural-to-mesenchymal transition, with the precise instructions of using "*Tabula Muris*" as input and expand our "*Validation on a broader basket of cell types would significantly strengthen the findings of the paper and also enable a better assessment of how general and natural the method is to apply outside of situations the authors have studied in detail below.*" We have therefore addressed an important question in the field of gene therapy using AAV vectors: can LSD design short and potent promoters that fit the size constraints of AAV vectors? We have clearly stated in the title of said paragraph that "*LSD enables designing of sLCRs compatible with size constraints of AAV-vectors.*" Hence, our data clearly support that LSD may be used to address questions outside the "comfort zone" of GBM PMT and - simultaneously - offer experimental validation in "*a broader basket of cell types*".

We are not entirely sure of what the Reviewer means by "bait-and-switch" because we have formulated a scientific problem to address (bait?) but we did not "switch" from the task we were assigned (i.e. different from GBM PMT, tissue-specific scRNA-seq inputs, validated in multiple cell lines from different tissues). Moreover, we are sure that the mammalian regulatory logic has not been solved to date. Hence, LSD being able to generate fully functional promoters in 100% of the cases tested so far, including one *de novo* guided by our R2 during this revision, in a field that is clearly not ours, is an important support to the general use of LSD. This is in our view a way to directly address the point raised by R2 and hope that this Reviewer will also appreciate our response to the critique now that we had the chance to elaborate further.

We will address further the matter of the specificity below.

The authors seem totally capable of making and experimentally testing the same number of reporters in the same number of cell lines, not housekeeping but cell-type-specific ones. Why they chose not to do so is beyond me.

We are happy to explain our reasoning, so that it's not left to imagination.

First, we acknowledge that this Reviewer might not have had the chance to see our response to R2 and therefore their piggyback critique may be based on the first set of comments (i.e. Nov 2022). We have addressed the point that LSD can design specific sLCRs in the most robust possible ways prior to submission. Specifically:

- 1) we have shown that LSD can design functional and specific sLCRs that perform equally well or better than 1st. Gen. ones (see our response above and the manuscript in general).
- 2) we have used LSD to design sLCRs for SARS-CoV-2 responses in epithelial cells and published this during the rebuttal period (Jiang, Schmitt et al. 2023 Science Adv.). We had attached this manuscript to the current one for Reviewers to be reassured that we had addressed this important point before submission. We realize that this might have been overlooked because of the large amount of info our Reviewers had to access (or glitch in the system, we have no way to tell which one is which). The paper references LSD as the source of the design and is now online ([DOI: 10.1126/sciadv.adf4975](https://doi.org/10.1126/sciadv.adf4975)). By definition, the latter addressed the concern "*how well their method performs beyond glioblastoma.*"

Nevertheless, we appreciated the challenge by our R2 as an invitation to make the specific case of how to use LSD in a totally different field. We decided to focus on the above-mentioned question in gene therapy (i.e. can LSD design short and potent promoters that fit the size constraints of AAV vectors?). Being non-expert in the field of gene therapy, which was also an important point of the said challenge, we have assessed that AAV-based gene therapy is generally achieved by modulating the specificity of the vectors using two orthogonal strategies. First, AAVs have tissue tropism that can be modulated by serotype selection and capsid engineering. Second, promoter selection and tissue-specific enhancers should ensure cargo expression once the AAV has reached the target. LSD can help to address the latter problem by:

- 1) ensuring broad expression in every tissue (e.g. housekeeper-like)
- 2) enhancing on-target expression in those tissues for which serotype selection and capsid engineering are insufficiently specific (e.g. tissue-specific like).

We were gifted by R2 also wearing their "author's hat" and recognizing in their first comment that "*Clearly expression and transduction and access [are] important constraints. I think a reasonable validation would involve 4-5 cell-types that span a set of different underlying tissue types.*" We did very much welcome this reasonable approach and thanked R2 in our rebuttal. Designing *in vivo*-specific AAVs would only be attained *in vivo*, which is not possible in the framework of a reasonable revision, and would address an unmet need in biotechnology that deserves its own space. We have started doing this in collaboration with experts, but we know it will take years to do it in a way that is sufficiently robust.

Needless to say, we could have decided to design an sLCR that would only work in one cancer cell line rather than the other, as this Reviewer seems to imply we should have had. However, since we were offered the opportunity to address an exciting biotech question, we felt that creating promoters that fit AAV size constraints would be more interesting for the broad readership of Nat. Comm. and we enthusiastically agreed that this was the most appropriate course of action. We hope that this Reviewer will wear their author's hat as well and acknowledge that our choice made in the absence of a specific request is reasonable and our data do address the question we decided to ask.

Reviewer #3 (Remarks to the Author):

I thank the authors for the amount of work and patient explanations. One thing I could deduce from their sometimes impassioned response is an oversight on my part: when I say things like "MGT4 is better than MGT1", I implicitly meant "experimentally" better, but given the authors' computational perspective, it might have left them the impression that I failed to acknowledge their computational analyses. This is one of several examples, so I apologize for not explicitly saying "experimental" when proper. Regardless, their rebuttal data and other published work provide strong evidence for the utility of their approach, and I support the publication of the manuscript.

Reviewer #4 (Remarks to the Author):

The authors have improved significantly the manuscript to address the reviewers' comments. I agree with the previous reviewer that Figure 5 is one of the most important results; the newly updated text and description help better understand the message. The results presented are scientifically sound and show the possibility of using the framework developed by the authors to deliver ubiquitous expression.

Nevertheless, I am confused about why the authors chose to design and test synthetic regulatory DNA to deliver housekeeping functionality. Indeed, the most critical and difficult aspect of gene therapy delivery using AAVs is to obtain cell-type specific expression (for instance, in specific neuronal types), which is not tested here. While one can hope that the framework could provide such property, it is not fully supported by the data yet. The authors should at least discuss this.

Point-by-point response to the reviewers' comments

Reviewer #3 (Remarks to the Author):

I thank the authors for the amount of work and patient explanations. One thing I could deduce from their sometimes impassioned response is an oversight on my part: when I say things like "MGT4 is better than MGT1", I implicitly meant "experimentally" better, but given the authors' computational perspective, it might have left them the impression that I failed to acknowledge their computational analyses. This is one of several examples, so I apologize for not explicitly saying "experimental" when proper. Regardless, their rebuttal data and other published work provide strong evidence for the utility of their approach, and I support the publication of the manuscript.

GG: We appreciated the engagement of this Reviewer and the opportunity to address their critiques. We believe to have addressed the matter of computational vs experimental performance in our earlier response. Here, we would like to acknowledge that we generally recognize the importance of clarifying our interpretations of the data and of earning Reviewers' support.

Reviewer #4 (Remarks to the Author):

The authors have improved significantly the manuscript to address the reviewers' comments. I agree with the previous reviewer that Figure 5 is one of the most important results; the newly updated text and description help better understand the message. The results presented are scientifically sound and show the possibility of using the framework developed by the authors to deliver ubiquitous expression.

Nevertheless, I am confused about why the authors chose to design and test synthetic regulatory DNA to deliver housekeeping functionality. Indeed, the most critical and difficult aspect of gene therapy delivery using AAVs is to obtain cell-type specific expression (for instance, in specific neuronal types), which is not tested here. While one can hope that the framework could provide such property, it is not fully supported by the data yet. The authors should at least discuss this.

GG: We are grateful to this Reviewer for the acknowledgment of the improvements made to the manuscript and the recognition of the significance of Figure 5 in supporting the main conclusion. The confusion expressed regarding the focus on synthetic regulatory DNA for housekeeping functionality rather than cell-type specific expression is duly noted. In response, we would like to re-emphasize that our primary goal during the revision of this manuscript was to address the challenge of AAV size, a more attainable goal during the revision of our computational framework. We acknowledge the importance of cell-type specific expression in gene therapy and recognize that while our framework lays the groundwork for such applications, it requires dedicated investigation, resources and in vivo validation. We have further modified the discussion to hopefully more explicitly state both the breadth of LSD' validation (mesenchymal glioblastoma, cancer EMT, SARS-CoV-2, functionality without minimal promoters, small size functional promoters) as well as the current limitations in our ability to state that cell-type specificity is systematically attainable

and the need for future studies to explore this aspect in depth. For simplicity, here we reproduce the relevant passage of the discussion:

“[...] However, achieving optimal transgene expression and cell-type specificity remains a key challenge in vector design, requiring precise delivery to target cells while avoiding unintended effects in non-target tissues. The use of HK promoters is sufficient in cases where targeted delivery is attainable, while in all other cases, tissue-specific cis-regulatory elements will be necessary. We demonstrated that designing tissue-specific sLCRs provides specificity in silico, but in vivo translation will necessitate specific tissue focus, deep gene expression, and chromatin accessibility datasets, along with attention to in vivo delivery. LSD's systematic success in designing functional promoters suggests its potential as a blueprint for developing tissue-specific promoters, further enhanced by cell-type-specific TFBS, which has proven effective in focusing on specific transcriptional programs⁵⁰⁻⁵³.”

Finally, we would like to thank the Reviewer for their constructive feedback and the tie-breaking role they played in supporting the publication of this manuscript in *Nature Communications*.